# The association of UBAP2L and G3BP1 mediated by small nucleolar RNA is essential for stress granule formation

Eri Asano-Inami [1,2✉], Akira Yokoi [1,3✉], Mai Sugiyama[2], Toshinori Hyodo[4], Tomonari Hamaguchi[5] & Hiroaki Kajiyama[1]

Stress granules (SGs) are dynamic, non-membranous structures composed of non-translating mRNAs and various proteins and play critical roles in cell survival under stressed conditions. Extensive proteomics analyses have been performed to identify proteins in SGs; however, the molecular functions of these components in SG formation remain unclear. In this report, we show that ubiquitin-associated protein 2-like (UBAP2L) is a crucial component of SGs. UBAP2L localized to SGs in response to various stresses, and its depletion significantly suppressed SG organization. Proteomics and RNA sequencing analyses found that UBAP2L formed a protein-RNA complex with Ras-GTP-activating protein SH3 domain binding protein 1 (G3BP1) and small nucleolar RNAs (snoRNAs). In vitro binding analysis demonstrated that snoRNAs were required for UBAP2L association with G3BP1. In addition, decreased expression of snoRNAs reduced the interaction between UBAP2L and G3BP1 and suppressed SG formation. Our results reveal a critical role of SG component, the UBAP2L/ snoRNA/G3BP1 protein-RNA complex, and provide new insights into the regulation of SG assembly.

[1] Department of Obstetrics and Gynecology, Nagoya University Graduate School of Medicine, 65 Tsuruma-cho, Showa-ku Nagoya 466-8550, Japan. [2] Bell Research Center for Reproductive Health and Cancer, Department of Obstetrics and Gynecology, Nagoya University Graduate School of Medicine, 65 Tsuruma-cho, Showa-ku Nagoya 466-8550, Japan. [3] Institute for Advanced Research, Nagoya University, Nagoya, Japan. [4] Department of Biochemistry, Aichi Medical University School of Medicine, Nagakute, Aichi 480-1195, Japan. [5] Division of Neurogenetics, Center for Neurological Diseases and Cancer, Nagoya University Graduate School of Medicine, 65 Tsuruma-cho, Showa-ku Nagoya 466-8550, Japan. ✉email: e-inami@med.nagoya-u.ac.jp; ayokoi@med.nagoya-u.ac.jp

Messenger ribonucleoprotein (mRNP) granules are non-membranous dynamic structures that contain non-translating messenger RNAs (mRNA) and proteins that regulate various stages of the life cycle of cellular mRNAs, including mRNA translation, pre-mRNA processing, localization, and degradation[1]. The major types of mRNP granules are stress granules (SGs)[1,2], P-bodies (PBs)[1,3], germ granules[4], and neuronal transport granules[5]. Among these, SGs and PBs are the most well-characterized mRNP granules in cells. SGs are absent under normal growth conditions but are induced by stress, such as heat shock, osmotic shock, or oxidative stress. PBs are usually observed under normal growth conditions and are involved in RNA decay[1]. Both structures share many of the same proteins and RNAs, and both are dynamically fused and separated under stressed conditions[6]. SGs have been reported to be associated with the pathogenesis of cancer, neurodegeneration, inflammatory disorders and viral infections[7]; however, the details of SG assembly/disassembly and the resulting effects on cell signaling and survival programs remain unknown.

The stresses that induce SG organization activate one or two of four protein kinases, HRI (heme-regulated initiation factor 2 α kinase), PERK (PKR like endoplasmic reticulum kinase), PKR (protein kinase R), and GCN2 (general control nonderepressible 2). These kinases phosphorylate serine 51 of eIF2α, which inhibits formation of the eIF2/tRNAi$^{Met}$/GTP ternary complex that is essential for translation initiation[8]. In the absence of this ternary complex, translating ribosomes are unable to function, leading to the accumulation of translationally stalled mRNPs. Some mRNP components are phosphorylated, ubiquitinated[9,10], arginine methylated[11–13], or O-GlcNAc modified[14] under stressed conditions, promoting mRNPs aggregations for SG organization. Although eIF2α phosphorylation is a major trigger for SG assembly, some chemicals have been reported to induce SG formation independent of phosphorylation, through the disruption of eIF4E interaction with eIF4G[15].

SGs are composed of polyadenylated mRNAs, translation initiation factors (eukaryotic initiation factors; eIFs), the 40 S ribosomal subunit, and various RNA binding proteins (RBPs) that control mRNA structure and function[1,2]. One of the best studied RBPs in SGs is G3BP1 (Ras-GTP-activating protein SH3 domain binding protein 1), which is expressed in various tissues and conserved in a wide range of species. G3BP1 has a conserved acidic domain, a nuclear transport factor 2-like domain, several SH3 domain binding motifs, an RNA recognition motif, and an RGG/RG motif that is rich in arginine and glycine[16]. G3BP1 interacts with many SG components and possesses DNA/RNA helicase activity in vitro[17]. Ser 149 of G3BP1 is dephosphorylated under stressed conditions, which is critical for self-aggregation and SG nucleation[18,19]. G3BP2, a homolog of G3BP1, is also expressed in many tissues and cell lines, and depletion of both proteins inhibited SG formation[20]. SGs also contain other RBPs, such as TDP-43, FUS, TIA1, and ATXN2, whose loss of function may lead to neurological and neuromuscular disorders[21–24]. In addition to RBPs, SGs recruit multiple enzymes, including protein kinases, phosphatases, and methyltransferases, to alter signaling pathways during the cellular adaptation to stress[9–13].

A recent mass spectrometry analysis of purified SGs revealed that SGs contain various RBPs[25]. We recently identified that ubiquitin-associated protein 2-like (UBAP2L) is a novel substrate of protein arginine methyltransferase 1 (PRMT1) and is necessary to ensure proper kinetochore-microtubule attachment for the progression of mitosis[26]. Other reports have shown that UBAP2L is a BMI-binding protein necessary for cell survival[27] and is associated with cancer progression[28,29]. UBAP2L has an RGG/RG motif that is often found in RBPs[30]. Recent evidence suggests that UBAP2L plays an important role in SGs[31–34]; but, the exact function of the protein in cells is unknown. In this report, we show that UBAP2L is localized to SGs and is required for SG assembly. In addition, UBAP2L forms a complex with small nucleolar RNAs (snoRNAs) and G3BP1.

## Results

**UBAP2L localizes to SGs**. To determine whether UBAP2L localizes to SGs, HeLa cells treated with 0.5 mM arsenite or 0.3 M sorbitol for 30 min were immunostained for UBAP2L together with SG marker proteins, such as G3BP1, TIAR, eIF4E, and PABPC1. As shown in Fig. 1a, UBAP2L accumulated into multiple small dot-like structures and colocalized with each of these marker proteins 30 min after arsenite or sorbitol treatment, as in previous reports[31–34]. Other stresses that induce SG formation, such as heat and hydrogen peroxide treatment, also promoted colocalization of UBAP2L with G3BP1 (Supplementary Fig. 1a). Some SG-localized proteins are known to aggregate in cells when exogenously overexpressed. Consistent with this, GFP-tagged UBAP2L (GFP-UBAP2L) aggregated around the nucleus with endogenous G3BP1 or eIF4E under non-stressed conditions. Similar to endogenous UBAP2L, GFP-UBAP2L accumulated in dot-like structures after 30 min of arsenite treatment (Fig. 1b and Supplementary Fig. 1b). G3BP1 overexpression has been reported to induce eIF2α phosphorylation[35], which inhibits translation initiation to promote SG organization. However, GFP-UBAP2L expression did not affect eIF2α phosphorylation (Supplementary Fig. 1c). Cycloheximide (CHX) blocks SG formation by inhibiting translation elongation[36]. To further confirm UBAP2L localization to SGs, cells were treated with arsenite in the presence or absence of CHX and immunostained for UBAP2L and G3BP1. In the presence of CHX, UBAP2L, and G3BP1 localized diffusely throughout the cell and did not show any accumulation in SGs (Fig. 1c). These results clearly show that UBAP2L is an essential component of SGs. Some SG components are known to localize to PBs as well. Cells were immunostained for UBAP2L together with DCP1A, PB marker protein[1]. Although UBAP2L did not accumulate at DCP1A-positive dots, some UBAP2L positive dots were bordered by DCP1A-positive dots and merged to a lesser extent. (Fig. 1d). According to previous studies, the localization of UBAP2L to PB is more clearly visible in KO-G3BP1 and 2 cells[37], so we investigated whether UBAP2L localized to PBs in siR-NAG3BP1 and 2 double transfected HeLa cells (Supplementary Fig. 1e). The number of SG decreases when G3BP1 and 2 are knocked down with siRNA (Supplementary Fig. 1f), but p-body formation is unaffected (Supplementary Fig. 1g).

A slight increase in UBAP2L localization to the PB was observed after double knockdown of G3BP1 and 2. (Supplementary Fig. 1d, h). As a result, it is proposed that UBAP2L may also work with the PBs.

**UBAP2L is essential for SG assembly**. We next tested whether UBAP2L was required for SGs organization. HeLa cells transfected with two different siRNAs were treated with arsenite and immunostained for UBAP2L and G3BP1. As shown in Fig. 2a, transfection of UBAP2L siRNAs clearly disrupted SG organization. This result is consistent with previous reports[33,34]. Both siRNAs significantly reduced UBAP2L abundance, but the expression of other SG marker proteins, including G3BP1, TIAR, eIF4E, and PABPC1, was not affected (Fig. 2b). The disruption of SG formation can result from the inhibition of stress-induced eIF2α phosphorylation. To rule out this possibility, we examined eIF2α phosphorylation and translation inhibition after arsenite treatment in the absence of endogenous UBAP2L. UBAP2L depletion did not reduce arsenite-induced eIF2α phosphorylation (Fig. 2b). Consistent with this, translation was suppressed by

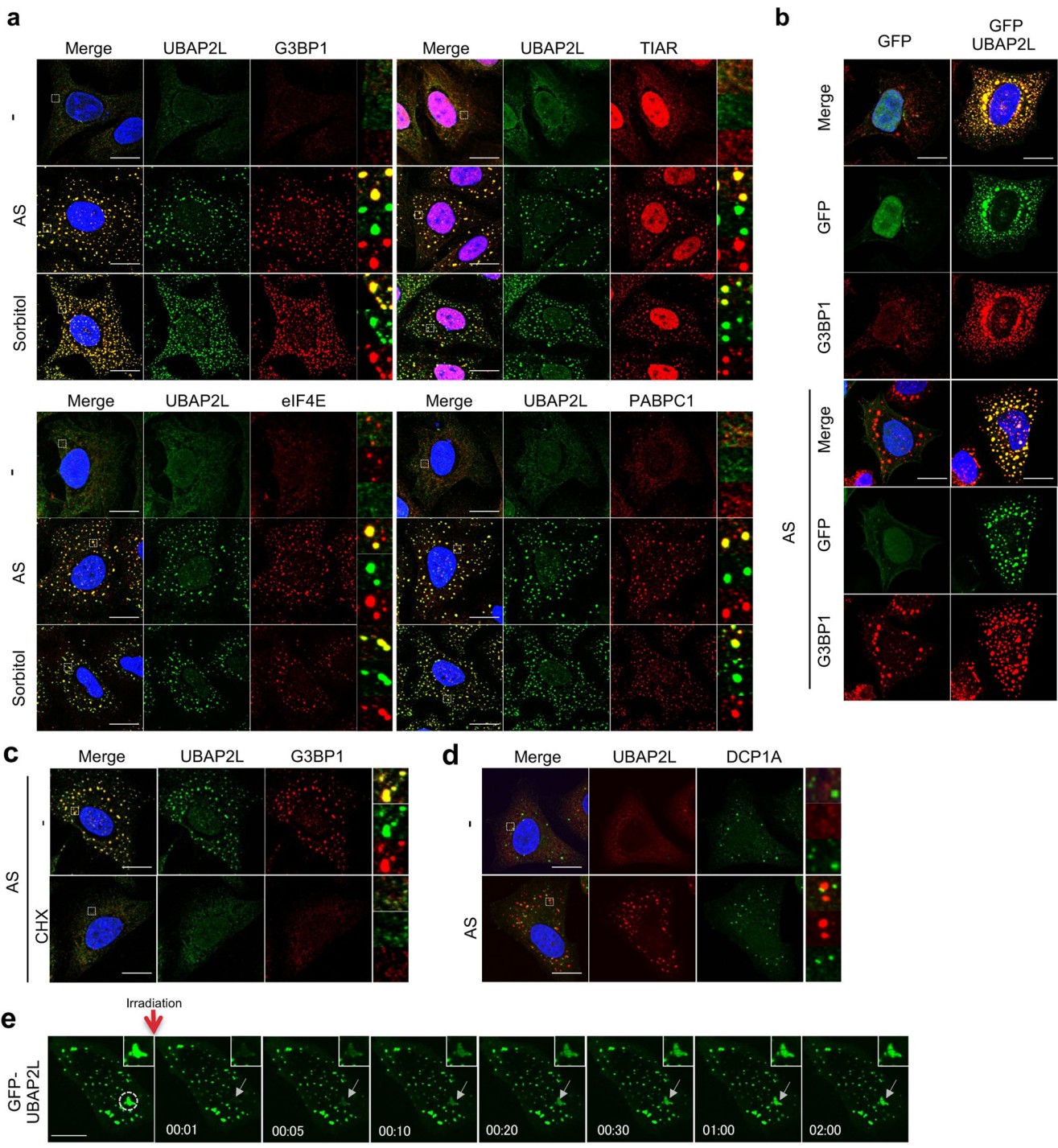

**Fig. 1 UBAP2L is localized to SGs. a** HeLa cells treated with 0.5 mM arsenite (AS) for 30 min or 0.3 M sorbitol for 30 min were immunostained for UBAP2L together with G3BP1, TIAR, eIF4E, or PABPC1 (scale bar = 10 μm). **b** HeLa cells were transfected with plasmids encoding GFP or GFP-UBAP2L, and 24 h later, the cells were treated with or without 0.5 mM arsenite (AS) for 30 min and immunostained for GFP and G3BP1(scale bar = 10 μm). **c** HeLa cells were treated with 0.5 mM arsenite for 30 min in the presence or absence of cycloheximide (CHX) and immunostained for G3BP1 and UBAP2L (scale bar = 10 μm). **d** Cells treated with 0.5 mM arsenite were immunostained for DCP1A and UBAP2L (scale bar = 10 μm). **e** Fluorescence photo bleaching recovery of a GFP-UBAP2L-expressing SG was monitored by confocal microscopy in the presence of 0.5 mM arsenite. The arrow shows the monitored SG. The time after irradiation is shown in each photograph (scale bar = 10 μm).

arsenite in UBAP2L-siRNA-transfected cells at a level similar to that of control-siRNA-transfected cells (Fig. 2c). To further confirm the critical role of UBAP2L in SG organization, UBAP2L-depleted cells were immunostained for other SG marker proteins (Fig. 2d), and the number of SGs per cell was counted. As shown in Fig. 2e-h, the number of SGs in UBAP2L-depleted cells was approximately 10-20% of that in control-siRNA-transfected cells. SG formation by heat shock and 0.3 M sorbitol stimulation was also reduced in UBAP2L-siRNA-transfected cells (Supplementary Fig. 2a, b). These results show that UBAP2L is essential for SG formation, independent of eIF2α phosphorylation and translation inhibition.

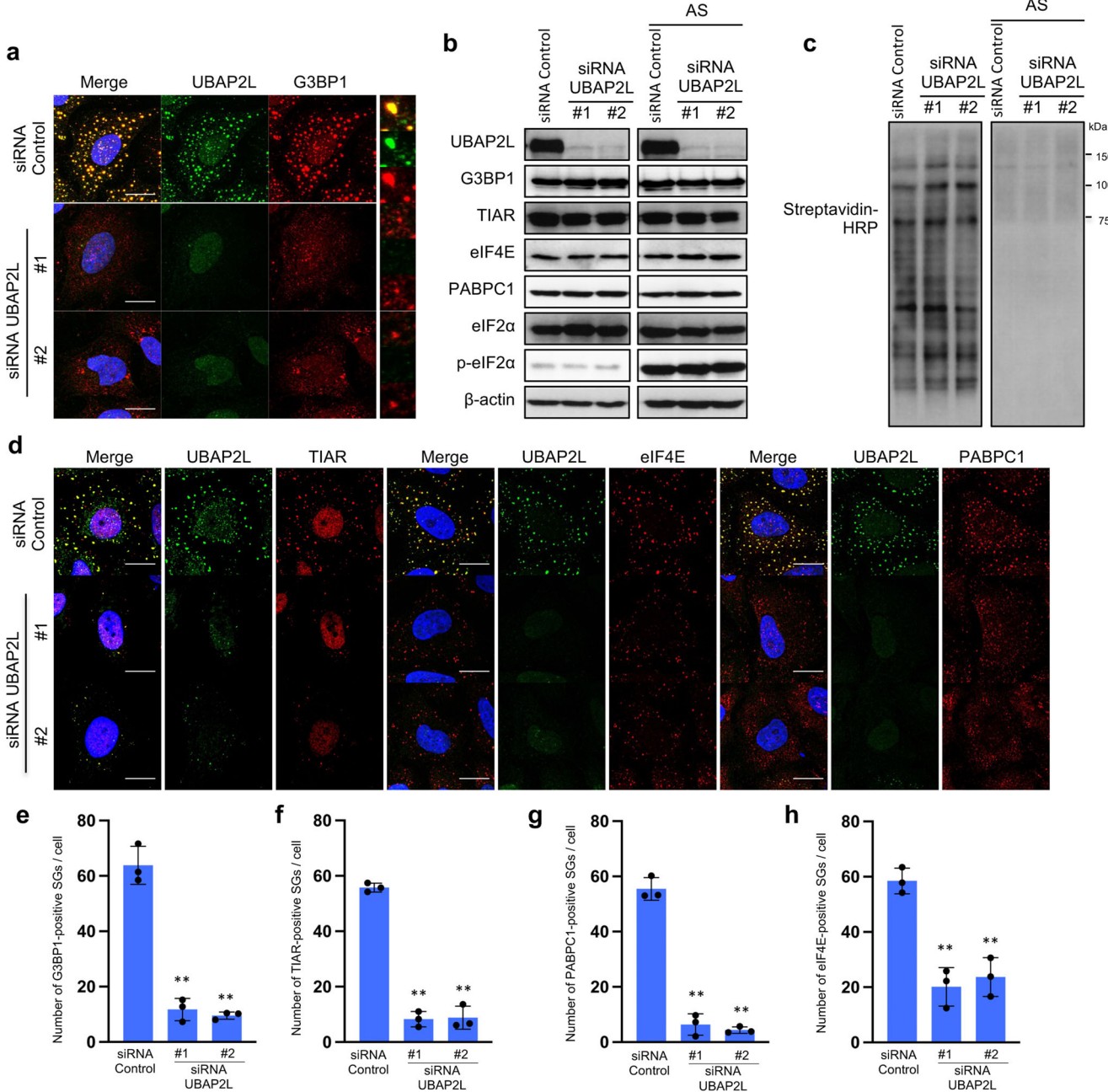

**Fig. 2 UBAP2L is essential for SG assembly. a** HeLa cells were transfected with control or UBAP2L siRNAs, and 72 h later, the cells were treated with 0.5 mM arsenite for 30 min and immunostained with anti-UBAP2L and anti-G3BP1 antibodies (scale bar = 10 μm). **b** Cells were transfected with the indicated siRNAs, and 72 h later, cells were treated with or without 0.5 mM arsenite for 30 min and lysed. The expression of UBAP2L, G3BP1, TIAR, eIF4E, or PABPC1 was examined by immunoblot. **c** Cells were treated with L-azidohomoalanine (AHA) with or without arsenite for 30 min, and lysed. The lysates were reacted with a Click-it Biotin Protein Analysis Detection Kit according to manufacturer's protocols. The reacted samples were immunoblotted with an anti-streptavidin-HRP antibody. **d** Cells transfected with siRNAs were treated as in **a** and immunostained for TIAR, eIF4E, or PABPC1 and UBAP2L (scale bar = 10 μm). **e–h** The numbers of SGs positive for each indicated SG marker per cell are presented in the graph. Three independent experiments were performed and 5 cells were evaluated for each experiment. (**P < 0.01).

**Amino acids 194-983 of UBAP2L are required for binding to G3BP1 and SG organization**. We performed mass spectrometry analysis to search for proteins that associate with UBAP2L. In this analysis, G3BP1/2, Caprin1, and the fragile X mental retardation family proteins; FXR1/2 and FMRP, which are SG-localized proteins[18–20,33,34,38], were found to coprecipitate with UBAP2L (Fig. 3a). To confirm the association of UBAP2L with these proteins, GFP-tagged G3BP1/2, FXR1/2, FMRP, or Caprin1 was transiently expressed in 293T cells together with FLAG-UBAP2L or FLAG tag.

The cell lysates were immunoprecipitated with an anti-FLAG antibody, and the immunoprecipitates were subjected to immunoblot analysis. GFP-G3BP1/2, FXR1/2, FMRP, and Caprin1 coimmunoprecipitated with FLAG-UBAP2L but not with the FLAG tag (Fig. 3b and Supplementary Fig. 3a). Endogenous UBAP2L also coprecipitated with these proteins (Fig. 3c and Supplementary Fig. 3b), indicating that UBAP2L associates with multiple SG-localized proteins.

UBAP2L has a ubiquitin-associated domain (aa50-80), RGG/RG motifs (aa134-189), the two putative RNA binding region

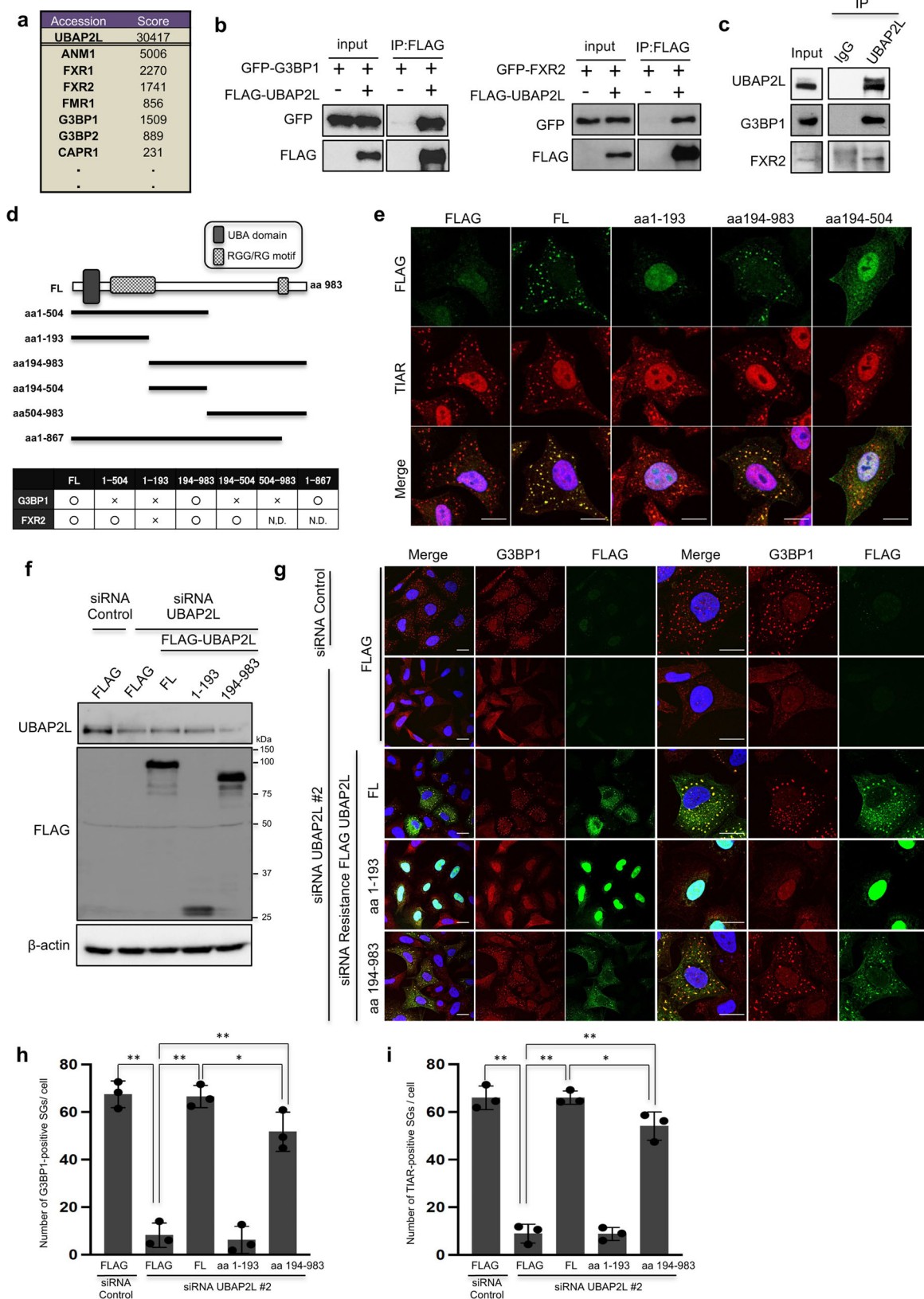

(aa239-290), and DUF (domain unknown function) motif (aa 495–526). To determine which regions are required for the association with SG-localized proteins, FLAG-tagged UBAP2L deletion constructs were transiently expressed in 293 T cells together with GFP-tagged G3BP1 or FXR2, and their association was examined by immunoprecipitation (Supplementary Fig. 3c, d).

We found that the aa194-983 fragment of UBAP2L, which contains DUF motif, was required for the interaction with G3BP1, in line with previous findings[34]. However, because the aa194-983 bond is weaker than that of FL, it is possible that aa1-193 also contains the region required for G3BP1 binding (Supplementary Fig. 3c). By contrast, the aa194-504 fragment

**Fig. 3 The interaction between UBAP2L and G3BP1 is required for SG assembly. a** List of proteins identified by mass spectrometry analysis. **b** 293T cells were transfected with FLAG-UBAP2L together with GFP-G3BP1 or GFP-FXR2. After 24 h, the cells were lysed and immunoprecipitated with an anti-FLAG antibody. The immunoprecipitates were immunoblotted for GFP and FLAG. **c** HeLa cells were lysed and immunoprecipitated with either anti-rabbit IgG or anti-UBAP2L antibodies. The immunoprecipitates were blotted with anti-UBAP2L, anti-G3BP1 or anti-FXR2 antibodies. **d** Schematic representation of UBAP2L deletion constructs (N.D. (not detected)). **e** HeLa cells that constitutively expressed each construct were treated with 0.5 mM arsenite for 30 min and immunostained for TIAR and FLAG (scale bar = 10 μm). **f** HeLa cells that constitutively expressed each siRNA resistant cDNA were depleted of endogenous UBAP2L by UBAP2L siRNA #2. The cells were lysed and immunoblotted with anti-FLAG and anti-UBAP2L antibodies. **g** Cells that constitutively expressed each constructs were depleted of UBAP2L by siRNA and immunostained for FLAG and G3BP1 (scale bar = 10 μm). **h** The number of G3BP1-positive SGs per cell is presented in the graphs (n = 3 (total cells = 15), **P < 0.01, *P < 0.05). **i** The number of TIAR-positive SGs per cell is presented in the graphs (n = 3 (total cells = 15), **P < 0.01, *P < 0.05).

was sufficient for the interaction with FXR2 (Fig. 3d). To assess whether these UBAP2L deletion mutants localized to SGs, HeLa cells constitutively expressing each construct were treated with arsenite and immunostained for the FLAG tag and TIAR. As shown in Fig. 3e, the aa194-983 fragment, which interacts with G3BP1, localized to SGs, whereas the aa1-193 fragment, which interacts with neither G3BP1 nor FXR2, did not. A region that is sufficient for the interaction with FXR2 (aa194-504) did not show any accumulation in SGs.

We examined whether the G3BP1-binding region of UBAP2L was essential for SG formation. We generated siRNA-resistant, FLAG-tagged full-length UBAP2L (FLAG-FL-UBAP2L(R)) and aa194-983 UBAP2L (FLAG-194-984(R)) by introducing silent mutations in a region targeted by UBAP2L siRNA #2. HeLa cells that constitutively expressed the FLAG tag alone, FLAG-FL-UBPA2L(R), FLAG-194-984(R), and FLAG-1-193 (FLAG-tagged aa1-193 UBAP2L) were transfected with UBAP2L siRNA #2 (Fig. 3f) and immunostained for FLAG together with G3BP1 or TIAR (Fig. 3g). Although FLAG-1-193 did not rescue SG organization, both FLAG-FL-UBAP2L(R) and FLAG-194-983(R) significantly restored the number of SGs produced per cell (Fig. 3h, i). These results reveal that the G3BP1 interacting region of UBAP2L is essential for SG assembly. However, when compared to FLAG-FL-UBAP2L(R), FLAG-194-984(R) has a slight but significant difference in the number of SGs (Fig. 3g), indicating that FLAG-1-193 region may also play a role in SG formation.

**Small RNAs regulate the interaction between UBAP2L and G3BP1.** Posttranslational modifications, such as phosphorylation and methylation, play a critical role in protein interactions. We previously reported that the N-terminal RGG/RG motif of UBAP2L was arginine methylated by PRMT1[26]; thus, we next examined whether arginine methylation was required for the interaction between UBAP2L and G3BP1. HeLa cells were treated with or without Adox, an arginine methylation inhibitor, and the interaction was examined by immunoprecipitation. As shown in Supplementary Fig. 4a, the addition of Adox did not affect the interaction. In addition, UBAP2L clearly accumulated in SGs in the presence of Adox (Supplementary Fig. 4b).

Mass spectrometry analysis showed that some serine/threonine residues of UBAP2L were phosphorylated. Cell lysates were immunoprecipitated with an anti-UBAP2L antibody and treated with or without lambda phosphatase (Supplementary Fig. 4c). The phosphatase treatment did not affect the interaction between UBAP2L and G3BP1, indicating that these modifications are not essential for the association of UBAP2L with G3BP1.

It has been reported that UBAP2L is an RBP[39]. G3BP1 is also known as an RBP[1–3], so we speculated that the interaction between UBAP2L and G3BP1 is regulated by RNA. HeLa cell lysates were immunoprecipitated with an anti-UBAP2L antibody and treated with Benzonase, which degrades all forms of DNA and RNA. Interestingly, the association between UBAP2L and

G3BP1 was diminished by Benzonase treatment (Fig. 4a), but the association between UBAP2L and FXR2 was not affected (Fig. 4a). We also used DNase I or RNase A to specifically degrade either DNA or RNA. The addition of DNase I did not affect the interaction (Fig. 4b) but RNase A treatment significantly reduced the interaction (Fig. 4c). These results imply that the association is dependent on the existence of RNA.

To determine if UBAP2L binds to RNA, we performed a RNA immunoprecipitation (RIP) assay. 293T cells that constitutively expressed the Halo tag or Halo-UBAP2L were immunoprecipitated, and large RNAs (>200 nt) or small RNAs (<200 nt) were isolated. Both large and small RNAs were specifically isolated from precipitates of Halo-UBAP2L-expressing cells (Fig. 4d). The Agilent Bioanalyzer was used to confirm the size and quality of each isolated RNA. These results show that UBAP2L is an RBP (Fig. 4e).

We examined whether large or small RNAs were required for the interaction between UBAP2L and G3BP1. FLAG-UBAP2L was transiently expressed in 293T cells and purified by immunoprecipitation using an anti-FLAG antibody, followed by extensive lysate with RIPA buffer with RNaseA (20 μg/ml) and washing with high-salt buffer to remove any binding proteins and RNAs. The purified FLAG-UBAP2L was incubated with GST, GST-G3BP1, or GST-FXR2, which were purified from bacteria, at 4 °C in the presence or absence of 20 μg of large RNAs or 8 μg of small RNAs. GST, GST-G3BP1, and GST-FXR2 were precipitated by glutathione agarose beads, and the precipitates were subjected to immunoblot with an anti-FLAG antibody. FLAG-UBAP2L associated with GST-FXR2 in the absence of RNA. By contrast, FLAG-UBAP2L was coprecipitated only in the presence of small RNAs (Fig. 4f). These results show that small RNAs are required for the interaction between UBAP2L and G3BP1 and that RNA is dispensable for the UBAP2L -FXR2 interaction.

**UBAP2L and G3BP1 form a complex with snoRNAs.** To determine which small RNAs mediate the interaction between UBAP2L and G3BP1, small RNAs isolated by RIP using Halo-UBAP2L and Halo-G3BP1 precipitates were analyzed by RNA sequencing. During analysis process, the method of reads assigned per million mapped reads (RPM) was adopted, and 14% were uniquely mapped to UCSC human genome 19. The RNAs with the top 10 RPM scores are shown in Fig. 5a and Supplementary Fig. 5a. A circle graph shows the classification of RNAs with the top 100 RPM scores (Fig. 5b and Supplementary Fig. 5b). Interestingly, 60–70% of the top 100 RNAs in each result were small nucleolar RNA (snoRNA). SnoRNA is approximately 60–300 nucleotides long and is necessary for modification of precursor ribosomal RNA (rRNA)[40]; however, there has been no report indicating a role for snoRNAs in SG organization. SnoR-NAs are classified into two groups, C/D box and H/ACA box snoRNAs. These two groups of RNAs have distinctive sequences, but some regions are evolutionarily conserved[41]. Most of the snoRNAs that we obtained by RNA sequencing analysis were C/D

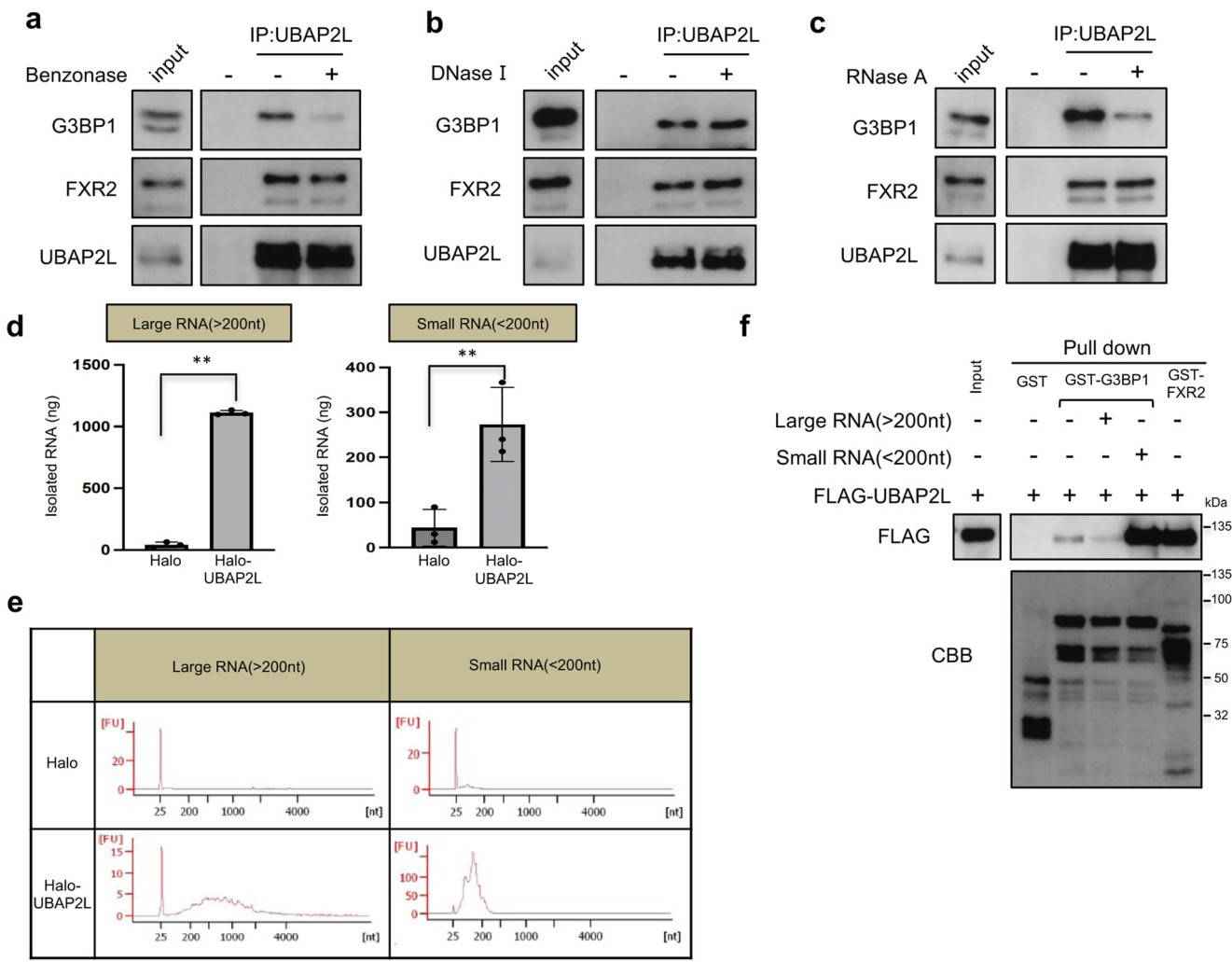

**Fig. 4 Small RNAs are required for the interaction between UBAP2L and G3BP1. a** UBAP2L was immunoprecipitated with an anti-UBAP2L antibody from HeLa cell lysates. The immunoprecipitates were treated with Benzonase and immunoblotted with anti-UBAP2L, anti-G3BP1, and anti-FXR2 antibodies. **b**, **c** HeLa cells were lysed and immunoprecipitated as in **a**. The immunoprecipitates were treated with DNase 1 or RNase A and immunoblotted with the indicated antibodies. **d** 293T cells that constitutively expressed the Halo tag or Halo-UBAP2L were immunoprecipitated with an anti-Halo antibody. Large RNAs (>200 nt) or small RNAs (<200 nt) were isolated from the immunoprecipitates. The graphs show the amounts of isolated RNAs (ng). Three independent experiments were performed (**$P$ < 0.01). **e** The Bioanalyzer quality analysis of each isolated RNA is presented in the graphs. **f** Immunoprecipitaed FLAG-UBAP2L was mixed with recombinant GST, GST-G3BP1, or GST-FXR2 bound to glutathione beads with or without larges RNAs (20 μg) or small RNAs (8 μg). The beads were precipitated and subjected to immunoblot analysis with an anti-FLAG antibody. CBB indicates Coomassie brilliant blue staining of recombinant proteins.

box snoRNAs (SNORDs) (Fig. 5a, b and Supplementary Fig. 5a, b). To confirm the association between snoRNAs and UBAP2L, the Halo tag, Halo-UBAP2L, Halo-G3BP1, or Halo-CPEB was expressed in 293T cells and precipitated using Halo beads. RNA was isolated from each precipitate and subjected to qRT-PCR analysis to detect SNORD44 and SNORD49A, which are C/D box snoRNAs. CPEB1 is an SG-localized protein that does not interact with UBAP2L. The analysis showed that both SNORD44 and SNORD49A bound to UBAP2L and G3BP1, but not CPEB1 (Fig. 5c). Furthermore, we carried out an in vitro binding assay using purified proteins and in vitro-transcribed SNORD44. Full-length SNORD44 was cloned into the pBluescript vector, and both the sense and anti-sense SNORD44 sequences were transcribed in vitro. In the presence SNORD44 sense sequence, FLAG-UBAP2L was clearly associated with GST-G3BP1 (Fig. 5d). By contrast, the anti-sense sequence of SNORD44 did not promote the interaction (Fig. 5d). We also examined other SNORDs, including SNORD49A, SNORD30, and SNORD56. The RPM

score of SNORD56 was significantly lower than those of SNORD30 and SNORD49A. As demonstrated in Fig. 5e, UBAP2L interacted with G3BP1 in the presence of these snoR-NAs; however, the SNORD56-mediated interaction was less significant than the SNORD30- or SNORD49A-mediated interaction. These results indicate that some specific C/D box snoRNAs form a complex with UBAP2L and G3BP1. Next, we purified the core fractionation of SG according to the paper conducted by Khong et al., 2017, Molecular Cell.,[42] and examined whether SNORD44 and SNORD49A existed there by qRT-PCR. As a result, it was found that these C/D box snoRNAs were present in the arsenic-treated SG core fraction more than in untreated HeLa cells (Fig. 5f and Supplementary Fig. 5c).

These results show that SNORD44 and SNORD49A were a slight but significantly detected in SG core fractionation. Since the previous report show that snoRNA was not found in the comprehensive RNA-seq analysis of the SG core fraction[42], snoRNA localized to SG may be very limited.

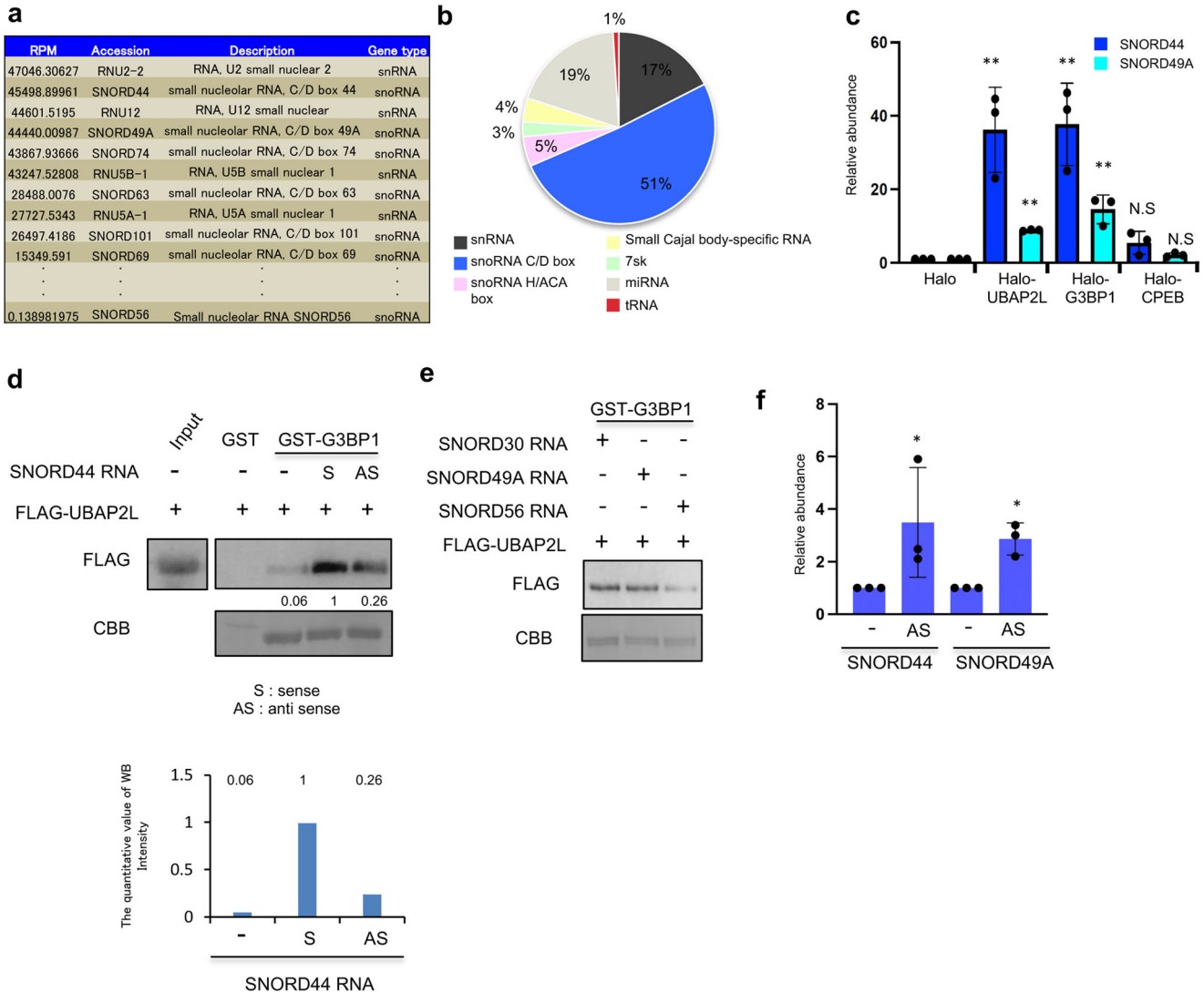

**Fig. 5 C/D box snoRNAs mediate the association between UBAP2L and G3BP1. a** The small RNAs with the top 10 UBAP2L RPM scores are presented (*n* = 2). **b** The percentages of each type of small RNAs with the top 100 UBAP2L RPM scores are shown in the circle graph (*n* = 2). **c** Small RNAs were isolated from immunoprecipitates from the lysate of 293T cells that constitutively expressed the Halo tag, Halo-UBAP2L, Halo-G3BP1, or Halo-CPEB. The levels of SNORD44 and SNORD49A were examined by quantitative RT-PCR (qRT-PCR) (*n* = 3, \*\**P* < 0.01, N.S. (not significant) *P* > 0.05). **d** Immunoprecipitated FLAG-UBAP2L was mixed with recombinant GST or GST-G3BP1 and incubated with or without in vitro-transcribed SNORD44 sense (S) or anti-sense (AS) sequences. Protein-RNA complex precipitated by glutathione agarose beads were immunoblotted with an anti-FLAG antibody. CBB indicates Coomassie brilliant blue staining of recombinant proteins. The number in the middle of the figure and the graphs indicates the quantitative value measured with ImageJ. **e** In vitro-transcribed SNORD30, SNORD49A, or SNORD56 were mixed with immunoprecipitated FLAG-UBAP2L and GST-G3BP1 and subjected to precipitation with glutathione agarose beads followed by immunoblot analysis. Both UBAP2L and G3BP1 RPM scores of SNORD56 is significantly lower than that of SNORD30 and SNORD49A. **f** SG core fraction was purified from arsenate treated HeLa cells and no-treated HeLa cells. RNA was purified each fraction. The levels of SNORD44 and SNORD49A were examined by qRT-PCR (*n* = 3, \**P* < 0.05).

**C/D box snoRNAs regulate SGs assembly**. We next investigated whether C/D box snoRNAs were required for SG formation. Most C/D box snoRNAs are processed from spliced introns to become functional for rRNA modifications. NOP56, NOP58, NHP2L1, and fibrillarin are critical factors in the generation of C/D box snoRNAs[43]. To reduce the abundance of C/D box snoRNAs, HeLa cells were transfected with NOP56 or NOP58 siRNA. Transfection of siRNAs significantly decreased NOP56 and NOP58 mRNA expression (Fig. 6a). The abundance of SNORD49A was significantly suppressed by transfection of either siRNA, whereas SNORD44 expression was reduced only by NOP58-depletion (Fig. 6b).

We next examined whether SG assembly was suppressed by either NOP56 or NOP58 depletion. SGs were organized in cells depleted of NOP56 or NOP58 (Fig. 6c); however, the number of SGs per cell in cells transfected with either NOP56 or NOP58 siRNA was significantly lower than that in control siRNA-transfected cells (Fig. 6d). Finally, we examined the association between UBAP2L and G3BP1 in cells transfected with both NOP56 and NOP58 siRNAs. HeLa cells transfected with both NOP56 siRNA #1 and NOP58 siRNA #1 were lysed and immunoprecipitated with an anti-UBAP2L antibody. As shown in Fig. 6e, the interaction between UBAP2L and G3BP1 was suppressed in NOP56- and NOP58-depleted cells. Collectively, these results show that C/D box snoRNAs mediate the association between UBAP2L and G3BP1 and promote SGs assembly under stress conditions. Using deletion mutants, we then determined which regions of UBAP2L are required for snoRNA binding.

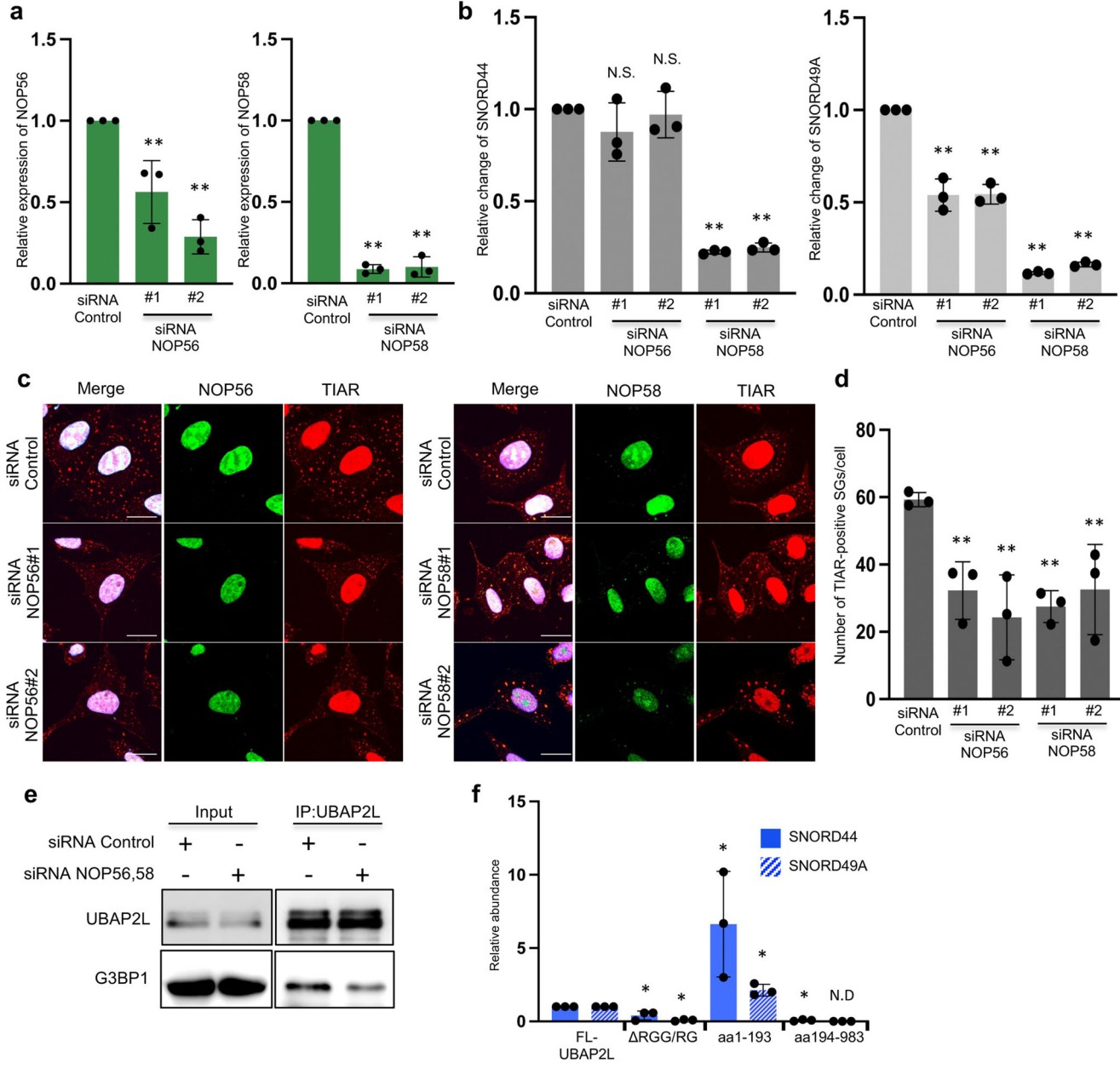

**Fig. 6 C/D box snoRNAs regulate SG assembly. a** HeLa cells were transfected with control, NOP56 or NOP58 siRNAs. After 72 h, the expression of NOP56 (left graph) or NOP58 (right graph) mRNA was examined by qRT-PCR. Three independent experiments were performed (**$P < 0.01$). **b** The abundance of SNORD44 (left graph) or SNORD 49 A (right graph) was examined by qRT-PCR ($n = 3$, **$P < 0.01$, N.S. (not significant) $P > 0.05$). **c** HeLa cells were transfected with control, NOP56, or NOP58 siRNAs, and 72 h later, the cells were treated with 0.5 mM arsenite for 30 min and immuostained with anti-TIAR together with anti-NOP56 or anit-NOP58 antibodies (scale bar = 10 μm). **d** The number of SGs per cell is presented in the graph. Three independent experiments were performed ($n = 20$, **$P < 0.01$). **e** HeLa cells were transfected with control siRNA or a combination of NOP56 #1 and NOP58 #1 siRNAs, and 72 h later, the cells were lysed and immunoprecipitated with anti-UBAP2L antibody. The immunoprecipitates were subjected to immunoblot analysis. **f** Small RNAs were isolated from immunoprecipitates from the lysate of 293T cells that constitutively expressed the FLAG-UBAP2L, FLAG-1-193, FLAG-ΔRGG/RG(130-193), or FLAG-194-983. The levels of SNORD44 and SNORD49A were examined by qRT-PCR ($n = 3$, *$P < 0.05$, N.D. (not detected)).

In 293T cells, FLAG-UBAP2L, FLAG-1-193, FLAG-ΔRGG/RG (130-193), or FLAG-194-983 was expressed and precipitated with FLAG beads. RNA was extracted from each precipitate and analyzed using qRT-PCR. As illustrated in Fig. 6f, binding to snoRNA was observed in the aa1-193 region. Furthermore, the absence of the RGG/RG region, which is an RNA-binding motif, reduces binding to snoRNA. It has been proposed that it is via the RGG/RG motif of UBAP2L.

## Discussion

In this report, we identified a crucial role for UBAP2L in SG organization. Endogenous as well as exogenously expressed GFP-UBAP2L clearly accumulated in SGs in response to various stresses. Knockdown of UBAP2L by two different siRNAs significantly reduced the number of SGs per cell, and exogenous expression of siRNA-resistant UBAP2L restored SG formation in cells depleted of endogenous UBAP2L. These results clearly show

that UBAP2L is a crucial component of SGs and essential for SG assembly. UBAP2L knockdown did not affect stress-induced eIF2α phosphorylation or translation inhibition, indicating that UBAP2L does not affect pathways leading to translational repression. In addition, expression of other SG-localized proteins was not reduced by UBAP2L depletion. Thus, it is likely that UBAP2L is required for SG assembly. Our mass spectrometry analysis further supported the importance of UBAP2L in SG organization. UBAP2L formed a complex with other SG-localized proteins, including FMR1, G3BP1, and Caprin1. UBAP2L is conserved in a wide range of species, and the Drosophila homolog is called Lingerer. A previous study reported that Lingerer interacted with FMR1, Caprin, and Rasputin, a Drosophila homolog of G3BP1[44]. Although whether Lingerer is required for SG formation remains to be investigated, it was shown that Lingerer regulates the JAK/STAT signaling pathway to restrict cell proliferation during development[45].

Despite mounting evidence that UBAP2L is required for SG formation[31–34], the functions of mammalian UBAP2L remain unknown. Recent studies showed that UBAP2L is associated with cancer cell proliferation[28,29]. Therefore, mammalian UBAP2L may be involved in processes other than SG formation, such as cell proliferation, by regulating signaling pathways.

Furthermore, UBAP2L is known to undergo methylation by PRMT1[26,34]. Huang C et al reported that Adox treatment increases SG formation[34]. Our results were not perfectly concordant with this report, and there might be complicated processes. Further analysis is needed to clarify this point.

Our RIP assay and in vitro binding assay revealed that UBAP2L association with G3BP1 was mediated by snoRNAs. There are two types of snoRNAs, C/D box snoRNAs and H/ACA box snoRNAs. Interestingly, most of the snoRNAs obtained by RNA sequencing were C/D box snoRNAs. Depletion of proteins that are required for the generation of mature C/D box snoRNAs significantly suppressed UBAP2L association with G3BP1 and SG formation. These results indicate that C/D box snoRNAs are critical for the assembly of the UBAP2L/snoRNA/G3BP1 protein-RNA complex to promote SG organization. The interaction of UBAP2L with other SG proteins (for example, FXR1/2 and Caprin) may also play a role in SG formation. Our findings show that RNA does not regulate the binding of UBAP2L and FXR2, and that Halo-CPEB (SG protein that is not bound to UBAP2L) does not bind to snoRNA. Because of many SG proteins bind to RNA, some other SG proteins may bind to snoRNA, but these findings indicate that the UBAP2L/snoRNA/G3BP1 protein-RNA complex is unique. C/D box snoRNAs are 60-100-nucleotide-long transcripts characterized by conserved box C (RUGAUGA, where R is a purine) and box D (CUGA) motifs[46]. The major function of C/D box snoRNAs is 2′-O-methylation of their target rRNAs; however, recent studies have revealed that C/D box snoRNAs have additional functions[47,48]. For example, SNORD60 is involved in intracellular cholesterol trafficking[49], and SNORDs U32a, U33, and U35a mediate lipotoxic stress, possibly through their cytosolic function[50]. Furthermore, SNORD55A and SNORD55B have been shown to bind and inhibit K-Ras and suppress tumorigenesis[51]. These reports and our study show that C/D box snoRNAs have multiple functions other than rRNA modification.

Accumulating studies have revealed that not only mRNAs but also miRNAs are involved in SG assembly[52,53]. The localization of mRNA to SGs has been shown by fluorescence in situ hybridization (FISH) using dT probes[23]; however, miRNA localization to SGs has not been confirmed. We detected snoRNA by qRT-PCR from SG core fractions (Fig. 5f and Supplementary Fig. 5c), so snoRNA may be present in SG; but, as shown in supplementary Figure 5d, we could not detect C/D box snoRNA localization to

SGs by FISH using multiple probes. One reason we could not observe snoRNA localization to SGs may be the methodological difficulties of FISH analysis of small RNA, such as insufficient sensitivity or inability of the probes to access their target snoRNAs. Another possibility is that snoRNAs are processed into shorter RNAs in SGs. It has been shown that snoRNAs are processed into shorter RNAs called sno-miRNAs, which are over 18 nt in length and functions are unknown[54–59]. SGs contain RNA endonucleases, such as Argonaute family protein[52,53], which may shorten snoRNAs in SGs and cause them to be undetectable by FISH. Immunoprecipitation analysis showed that the UBAP2L/snoRNA/G3BP1 complex is formed in the absence of stress, and recent SG RNA-seq analysis showed snoRNA is not abundant in SG[42]. thus, snoRNAs may be necessary for the initial phase of SG assembly and once SGs are formed, snoRNAs in the complex may be cleaved by RNA endonucleases for other functions. Although our study shows that snoRNAs are important for SG organization, further investigations are necessary to confirm snoRNA localization to SGs and understand the mechanisms of snoRNA-mediated SG assembly and cell survival.

In summary, we have shown that UBAP2L forms a protein-RNA complex with G3BP1 and snoRNAs and that this complex is essential for SG assembly (Fig. 7a, b). UBAP2L also associates with additional SG-localizing proteins, such as FMR1, FXR1, FXR2, and Caprin1. In addition, UBAP2L interacts with not only small RNAs but also mRNAs. RNA sequencing demonstrated that several mRNAs encoding apoptosis-related proteins are associated with UBAP2L; thus, future studies elucidating the physiological roles of UBAP2L will provide more insight into the molecular mechanisms and relevance of SGs and cell survival.

## Methods

**Cells**. HeLa and 293T cells were purchased from RIKEN BRC (Tsukuba, Japan) and cultured DMEM containing 10% FBS at 37 °C.

**Antibodies**. To generate an anti-UBAP2L antibody, a fragment including amino acid (aa) residues 513-660 of UBAP2L fused with glutathione S-transferase (GST) was produced in bacteria, and recombinant protein was purified by using glutathione agarose beads (Sigma-Aldrich, St. Louis, MO, USA). The protein was mixed with Freund's adjuvant (Sigma-Aldrich) and injected into a rabbit 4 times, every 2 weeks. To purify the anti-UBAP2L antibody, we used Hi-Trap N-hydroxysuccinimide (NHS)-activated HP columns (GE Healthcare Bio-Sciences), coupled with recombinant GST-UBAP2L (aa 513-660). Other antibodies were obtained from the following companies: anti-G3BP1 (611126), anti-TIAR (610352) and anti-FXR2 (61130) antibodies were obtained from BD Biosciences (San Jose, CA, USA); anti-G3BP1 (A301-033A) and anti-G3BP2 (A302-040A) antibodies were obtained from Bethyl Laboratories (Montgomery, TX); anti-eIF2α (5324), anti-phospho-eIF2α(3398), anti-FXR1 (12295),anti-FMRP (4317), and anti-streptavidin-HRP (3999) antibodies were obtained from Cell Signaling (Danvers, CA, USA); anti-eIF4E (sc-9976) and anti-PABPC1 (sc-32318) antibodies were obtained from Santa Cruz Biotechnology (Santa Cruz, CA, USA); anti-DCP1A antibody (H00055802-M06), Abnova (Taipei, Taiwan); anti-FLAG antibody (014-22383), Wako (Osaka, Japan); anti-GFP (598) and anti-DIG (M227-3) antibodies were obtained from MBL (Nagoya, Japan); anti-Halo antibody (G921A) was obtained from Promega (Madison, WI, USA); and then anti-ADMA antibody were obtained from Active Motif (Carlsbad, CA, USA).

**Immunoblot analysis**. Proteins were loaded onto a Polyacrylamide Gel (8–10%,) and electrotransferred (100 v, 30 mA). The proteins were transferred to a polyvinylidene difluoride membrane (Millipore). After blocking in 0.1% skim milk, the membranes were incubated for 1-12 h at room temperature or 4° with primary antibodies. Secondary antibodies (Mouse IgG HRP Linked Whole Ab, NA931, Cytiva, Tokyo, Japan or Anti-rabbit IgG, HRP-linked Antibody #7074, Cell Signaling), were used at a dilution of 1:2000 for 1 h at room temperature. The membrane was then exposed to Light captureII (ATTO, Tokyo, Japan).

**siRNA transfection**. The following control siRNA and siRNAs used to suppress UBAP2L, G3BP1/2, FXR2, and NOP56/58 expression were purchased from Thermo Fisher Scientific (Waltham, MA USA): Silencer® Select siRNA Control (4390843); and Silencer® Select Pre-designed siRNA UBAP2L #1(s19176) and #2(s230223), G3BP1 #1 (s19754) and #2 (s19755), G3BP2 #1 (s19206) and #2 (s19207), FXR2 #1 (s18243) and #2 (s18244), NOP56 #1 (s20642) and #2 (s20643),

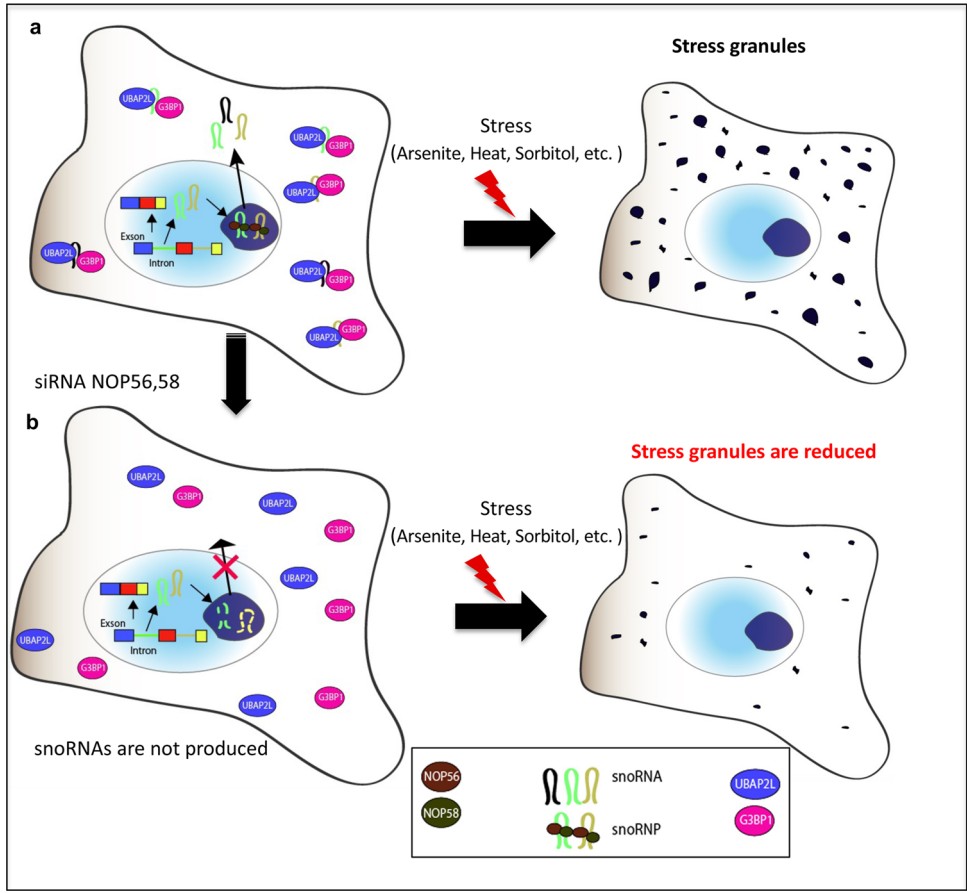

**Fig. 7 A schematic model of the UBAP2L-snoRNA complex in SG. a** In the nucleus, snoRNAs are produced by introns and complex with NOP56 and NOP58 as snoRNPs and matured by endonucleases. Some snoRNAs are produced in the cytoplasm. UBAP2L and G3BP1 complex with C/D box snoRNAs in the cytoplasm. Once cells are stimulated by stress, SGs are formed normally. The cells can adjust to the stress conditions to survive. **b** When the expression of C/D box snoRNAs is reduced by NOP56 or NOP58 depletion, UBAP2L and G3BP1 cannot form a complex. The number of SGs is reduced, and it is difficult for the cell to adjust to stress.

and NOP58 #1 (s28390) and #2 (s28391). HeLa cells were transfected with 5 nM siRNA using Lipofectamine RNAiMAX (Invitrogen, Carlsbad, CA. USA).

**Immunofluorescence analysis.** Cells cultured on fibronectin-coated glass cover slips were transfected with siRNAs or vectors and then stimulated with 0.5 mM arsenite for 30 min, 300 mM sorbitol for 30 min, 1 mM hydrogen peroxide for 1.5 h, or heat (44 °C) for 30 min. Cells were fixed with 4% paraformaldehyde for 10 min on ice and permeabilized with PBS containing 0.5% Triton for 5 min. The cells were blocked with PBS containing 7% FBS for 30 min and incubated with primary antibodies for 1 h. After washing with PBS, the cells were incubated with Alexa Fluor 488- or Alexa Fluor 594-labeled secondary antibodies (Invitrogen) and Hoechst (Dojindo) for 1 h. Images were acquired using an FV1000 laser scanning confocal microscope (Olympus, Tokyo, Japan).

**Fluorescence recovery after photobleaching.** Images were recorded using an FV1000 laser scanning confocal microscope (Olympus). HeLa cells constitutively expressing GFP-UBAP2L were cultured on a glass-bottom dish and treated with 0.5 mM arsenite for 20 min. An SG was irradiated with a laser 405 nm for 10 sec, images were scanned every second for 5 min after bleaching.

**RNA immunoprecipitation assay.** 293 T cells constitutively expressing the Halo tag, Halo-UBAP2L or Halo-G3BP1 were lysed in lysis buffer (25 mM Tris-HCl (pH 7.6), 150 mM NaCl, and 0.1% NP-40) supplemented with a phosphatase inhibitor cocktail (Nacalai Tesque, Inc., Japan), protease inhibitor (Promega), 1.5 mM DTT, and RNase inhibitor (Takara) for 10 min on ice and treated with DNaseI(Takara) for 10 min at room temperature. The lysates were centrifuged at 15,000 rpm for 10 min and the supernatants were incubated with Halo Resin (Promega) for 12 h. The resin beads were washed with wash buffer (25 mM Tris-HCl (pH 7.6), 150 mM NaCl, and 0.005% NP-40) five times, and suspended in ML buffer (NucleoSpin miRNA: Takara) to isolate large and small RNAs.

**Quantitative RT-PCR.** RNA was extracted from HeLa cells using the RNeasy Mini kit (Qiagen, Hilden, Germany), and cDNA was generated using PrimeScript reverse transcriptase (Takara). Small RNA was extracted from RIP assay using NucleoSpin miRNA (Takara), and cDNA was generated using miRCURY LNA™ Universal RT microRNA PCR Universal cDNA synthesis KitII (EXIQON, Copenhagen, Denmark). PCR was performed using the SYBR® Premix Ex *Taq*™ II (Takara) (for mRNA) or miRCURY LNA™ Universal RT microRNA PCR Exi-LENT SYBR® Green master mix (EXIQON) (for small RNA). The relative RNA expression levels were normalized to GAPDH or miRCURY LNA UniRT PCR Control primer mix UniSp6 (203954, EXIQON). The sequences of primers used to amplify each gene were: 5′-AGGTGGAGGAGTGGGTGTCGCTGTT-3′ and 5′-CCGGGAAACTGTGGCGTGATGG-3′ (GAPDH); 5′-CAGCATCCACAGTG CAGATC-3′and 5′-GCACCTCAGAGAAGCAATGC-3′(NOP56); 5′-TGGCAG CTATGTGTCTTGGA-3′and 5′-TGCCAGCCATACCATTCTCT-3′(NOP58); 5′- TGCTCTGATGAAATCACTAA-3′and 5′-AATCAGACAGGAGTAGTCTT-3′ (SNORD49A); miRCURY LNA UniRT PCR Reference primer mix SNORD44 (203902, EXIQON).

**Small RNA-sequencing.** Small RNAs isolated by RIP (Halo-UBAP2L and Halo-G3BP1) were tested using a Nanodrop 2000 instrument (Thermo Fisher Scientific) and RNA integrity was determined using an Agilent 2100 Bioanalyzer (Agilent, Japan). The NEB Next Small RNA Library Prep Set for Illumina was used to create the small RNA libraries (NEB). The HiSeq 1500 instrument was used for sequencing (Illumina). The sequencing reads were trimmed with the FASTX-Tool kit 0.0.13 and mapped to the human reference genome hg 19 (UCSC), and the RPM (reads per million mapped reads) was calculated with Strand NGS ver 2.1 (Agilent Technologies). The low RNA-seq data was displayed in the (Supple-mentary data 1).

**Statistical analysis.** Three independent experiments were performed, and the results were compared using Student's t-test. The data are represented as the means ± standard deviation (s.d). In the graphs of dot plot, the bars indicated

median, and statistical significance was determined by Unpaired t-Test. $P < 0.05$ was considered statistically significant.

**Reporting summary**. Further information on research design is available in the Nature Portfolio Reporting Summary linked to this article.

## Data availability

All other relevant data are available within the article file, Supplementary Figures or Supplementary data file, or available from the authors on reasonable request. Uncropped scans of the bolts were shown in Supplementary Fig. 6.

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

## Acknowledgements

We are grateful to Dr. Takeshi Senga for helpful discussions and comments on the manuscript. This study was financially supported through JSPS KAKENHI Grant Numbers 21H03075, Princess Takamatsu Cancer Research Fund, Daiichi Sankyo Foundation of Life Science, and the Uehara Memorial Foundation. Moreover, this study was also supported by Program for Promoting the Enhancement of Research Universities as young researcher units for the advancement of new and undeveloped fields at Nagoya University.

## Author contributions

E.A.I. designed, performed, and analyzed all experiments. M.S. helped with making cell lines and plasmids. T.Hyodo and T.Hamaguchi contributed data analysis. All authors were involved in the experiment design. A.Y. and H.K. provided helpful discussions. E.A.I. wrote the manuscript. A.Y. revised the manuscript.

## Competing interests

The authors declare no competing interests.
