## [Peer Review File · Communications Biology]

Response to reviewer comments

To Reviewer 1:

We appreciate your variable comments. All points were carefully considered, and our manuscript was revised to address your comments. Please see our point-by-point responses to your comment as shown below.

Referee #1 Comments for the Author...

In this manuscript the authors make two contributions. First, they demonstrate UBAP2L localizes to and promotes stress granule formation. Second, they argue UBAP2L promotes stress granule formation by forming a complex with snoRNAs and G3BP proteins. As detailed below, there are two major issues with the work that need to be addressed before the manuscript can be considered further.

Major concerns

(1) One issue is that prior results have demonstrated UBAP2L localizes to stress granules and is necessary for granule assembly (Cirillo et al., 2020 Current Biology; Youn et al., 2018 Molecular Cell; Markmiller et al., 2018, Cell). While it is fine for the authors to add data strengthening this conclusion, they need to acknowledge and reference the prior work.

►Author Response-1

Thank you for these suggestions, and we agree that the reports will help us in our work. The prior works were included in the revised manuscript as references, and the following sentences were added.

(Page 4, lines 110-111)

“Recent evidence suggests that UBAP2L plays an important role in SGs [31, 32 ,33]; but, the exact function of the protein in cells is unknown.”

(Page 11, lines 327-328)

“Despite mounting evidence that UBAP2L is required for SG formation [31, 32, 33], the functions of mammalian UBAP2L remain unknown.”

2) The second major issue is whether UBAP2L sufficient data is presented to argue that formation of a complex with snoRNAs and G3BP1 is how UBAP2L promotes stress granule formation. The limitations and possible manners to address them are:

a) The conclusion that small RNAs are necessary for interaction between UBAP2L and G3BP1 is primarily evidenced through figure 4f. However, the FLAG-UBAP2L input was immunoprecipitated

from cell lysate which allows for the possibility of an unidentified biomolecule contaminating the sample and affecting the results. Ideally, one would use purified UBAP2L (or a domain) for this pull down. Since it may be difficult to purify UBAP2L, this experiment would not be essential for publication but would greatly strengthen the manuscript.

►Author Response-2

We apologize for any confusion caused by a lack of explanation and agree with your concern that the FLAG-UBAP2L was immunoprecipitated from cell lysate, which allows for the possibility of an unidentified biomolecule contaminating the sample and influencing the results. To address this concern, we lysed the FLAG- UBAP2L expressing cells with RIPA buffer and RNase A (RNase A is usually used for protein binding experiments to reduce nonspecific RNA. Enomoto A et al.,2009 Neuron.) to reduce RNA and RNA binding proteins, and then washed out with high-salt buffer to remove any remaining nonspecific RNA. We assumed that these methods would remove the majority of nonspecific and specific binding proteins and RNAs. In the revised manuscript, we added the following detailed explanation.

(Page 8, lines 235-236)

“FLAG-UBAP2L was transiently expressed in 293T cells and purified by immunoprecipitation using an anti-FLAG antibody, followed by extensive lysate with RIPA buffer with RNaseA (20 μ g/ml) and washing with high-salt buffer to remove any binding proteins and RNAs.”

b) A second issue is that the distinction between small and large RNAs reconstituting the interaction is limited since this experiment used different amounts of input RNA (20 ug large RNAs and 8 ug small RNAs). I recommend using a constant mass of RNA.

►Author Response-3

We extracted large and small RNA from HeLa cells in 10 cm dish. The amount of large or small RNA obtained was approximately 20 μ g or 8 μ g, respectively, and this amount was used for pull-down assay (Figure 4f). To address your concerns, we ran additional experiments with a constant mass of 10 μ g of small or large RNA. As shown in the right Figure (for reviewers. 1), the results were identical to Figure 4f.

Figure for reviewers 1.

Immunoprecipitated FLAG-UBAP2L was mixed with recombinant GST or GST-G3BP1 bound to glutathione beads with or without large RNAs (10 µg) or small RNAs (10 µg). The beads were precipitated and subjected to immunoblot analysis with an anti-FLAG antibody. CBB indicates Coomassie brilliant blue staining of recombinant proteins.

c) A third issue is that the details of how their sequencing was standardized is not well explained. Did they compare the ip'd RNAs to a total RNA seq data set? This should be made explicit so that the validity of this enrichment can be fairly assessed.

►Author Response-4

We sincerely apologize for the lack of explanations. We investigated the IP Halo-UBAP2L and Halo-G3BP1 seq data sets. The low data of RNA-seq data was included in Supplementary data file 1 in the revised manuscript. The small RNA-sequencing method now includes these explanations.

(Page14, lines 440-447)

“Small RNAs isolated by RIP (Halo-UBAP2L and Halo-G3BP1) were tested using a Nanodrop 2000 instrument (Thermo Fisher Scientific) and RNA integrity was determined using an Agilent 2100 Bioanalyzer (Agilent, Japan). The NEB Next Small RNA Library Prep Set for Illumina was used to create the small RNA libraries (NEB). The HiSeq 1500 instrument was used for sequencing (Illumina). The sequencing reads were trimmed with the FASTX-Tool kit 0.0.13 and mapped to the human reference genome hg 19 (UCSC), and the RPM (reads per million mapped reads) was calculated with Strand NGS ver 2.1 (Agilent Technologies). The low RNA-seq data was displayed in the (Supplementary data file1)”

(4) Since the main conclusion is that UBAP2L and G3BP1 form a complex with snoRNAs that is essential for stress granule assembly, they should validate this result by providing evidence snoRNAs can be in stress granules. This is an issue since previous RNA-Seq data suggests snoRNAs are depleted from stress granules. Since the authors' FISH experiments did not show any snoRNAs in stress granules, I suggest the authors purify stress granules (see Khong et al., 2017, Molecular Cell) and perform RT-PCR for snoRNAs to see if snoRNAs can be detected in stress granules. Alternatively, the authors need to reduce the emphasis on this conclusion in the manuscript.

►Author Response-5

Thank you for insightful comment, we carried out the experiments you suggested. We purified a stress granule rich fraction from HeLa cells and performed RT-PCR for snoRNAs, as described in Khong et al., 2017, Molecular Cell. As a control, we extracted RNA from the SG rich fraction using an AS-untreated HeLa cell. The increased levels of SNORD44 and SNORD49A in the stress granule purified fraction were confirmed by RT-PCR (Figure 5f). However, since the difference is about twice that of the control, it is not considered to be so much. Since it could not be confirmed our FISH, snoRNA itself may not be present in stress granule. However, as shown in Figure 6, knockdown of NOP56 and 58 caused lower expression levels of SNORD44 and SNORD49A, the number stress granule decreases, and the reduction of binding between UBAP2L and G3BP1, and thus snoRNA may be involved in the early stages of granule formation. In the future, we hope to conduct a more detailed analysis and uncover more functional mechanisms of the snoRNA-UBAP2L-G3BP1 complex.

Figure 5f shows the results, and Supplementary figure 5c shows our experiment protocol. The revised manuscript now includes the following texts.

(Page 9, lines 281-283)

'Next, we purified the core fractionation of SG according to the paper conducted by Khong et al., 2017, Molecular Cell., [41] and examined whether SNORD44 and SNORD49A existed there by qRT-PCR. As a result, it was found that these C/D box snoRNAs were present in the arsenic-treated SG core fraction more than in untreated HeLa cells (Fig 5f and Supplementary Fig. 5c). These results shows that SNORD44 and SNORD49A were a slight but significant detected in SG core fractionation. Since previous report show that snoRNA was not found in the comprehensive RNA-seq analysis of the SG core fraction [41], SnoRNA localized to SG may be very limited.

Fig. 5f

Supplementary Fig. 5c

Minor concerns

5) Their FRAP experiment isn't very useful as it is presented. It would be more valuable if the FRAP was quantified by a graph and if they added some controls so that the reader can be aware of the significance of the result for the figure's conclusion.

►Author Response-6

Thank you for your thoughtful remarks. We only examined whether UBAP2L localization is fluid in this study, so we did not conduct a detailed analysis. In the future, we would like to thoroughly examine the points you raised

6) It would be interesting to know if they see any difference in the volumes of the stress granules in the Nop56 and Nop58 knockdowns.

►Author Response-7

Thank you for bringing this to our attention. NOP56 and NOP58 knockdown SG volumes were examined. Only in NOP58 knockdown was the SG volume 1.5–2 times greater than in control (Figure for reviewers 2). Because this result has no bearing on the conclusion, we did not include it in the text. It is very intriguing, and I would like to investigate it further in the future.

Figure for reviewers 2.

SG area was measured by Image J. 20 cells were evaluated for each experiment.

To Reviewer 2:

Thank you for your feedback on our manuscript. The manuscript was substantially revised in response to the reviewers. We hope that this revised version is better than the original version for you.

Referee #2 Comments for the Author...

In the manuscript, Inami et.al described the association of UBAP2L and G3BP1, and stated that UBAP2L, as well as its binding with G3BP1, is required for SG formation. They also showed that snoRNA is essential for their binding and SG formation.

However, the study is of little novelty and therefore has no significance. Chen's group (Huang C, Chen Y, Dai H, Zhang H, Xie M, Zhang H, Chen F, Kang X, Bai X, Chen Z. UBAP2L arginine methylation by PRMT1 modulates stress granule assembly. Cell Death Differ. 2020 Jan;27(1):227-241.) had described the role of UBAP2L in SG assembly, its association with G3BP1/2, as well as its regulation by PRMT1. Obviously, the authors missed this important literature. Therefore, the results showed in figures 1-3 in the manuscript had been known and the study is of very little novelty and significance. Besides, the manuscript also missed several other critical references with high relations, e.g. 1) UBAP2L Forms Distinct Cores that Act in Nucleating Stress Granules Upstream of G3BP1. 2020 Feb 24;30(4):698-707.e6. 2) Large-scale tethered function assays identify factors that regulate mRNA stability and translation. Nat Struct Mol Biol. 2020 Oct;27(10):989-1000.

Thus, the reviewer decline to accept the manuscript.

►Author Response-1

We agree that those reports are related and important to our work, and at the same time, they support the biology of UBAP2L is of interest to researchers working on cellular stress pathways and small RNA. As you listed, Recently, Luca C et al. *Curr.Biol.*2020 and Huang C et al. *Cell Death Differ.* 2020 reports show the possibility of the interaction between UBAP2L and SGs. However, these papers did not demonstrate how UBAP2L regulates SG formation and regulatory mechanisms between UBAP2L and SG component proteins. In this study we showed UBAP2L and G3BP1 bind via snoRNA and the complex is regulating SG formation and we believe that our data are also sufficiently novel and valuable findings. In the revised manuscript, those prior works were mentioned and included as references. In addition, based on the comments, we performed additional experiments, and have added two figures, six supplemental figures, and a supplemental data file. These assessments improved our works and we would appreciate if you recognize this revised manuscript as worthy report.

To Reviewer 3:

We would like to thank the reviewer for many insightful comments that helped us improve our manuscript. Please see the following for our point-by-point responses to your comment.

Referee #3: Comments for the Author...

The study focused on the role of UBAP2L protein in SG assembly and showed evidence that UBAP2L knockdown can reduce SG assembly. This conclusion is not novel but is worthwhile for independent validation. Another significant step forward for this study is trying to link the role of snoRNA in SG assembly, which I felt the evidence is weak. The only evidence for snoRNA interaction with UBAP2L is through in vitro pulldown. The pull-down could be unspecific, as shown by the author that there is even interaction without RNA addition. How abundant of snoRNA in SG? The author failed to detect snoRNA inside SG, and snoRNA is small RNA and was previously reported to be depleted inside the SG core structure. Below are a few comments for the author's consideration:

1. Use the gene names *UBAP2L* and *G3BBP1* in the title.

►Author Response-1

Thank you for your suggestions. The title has been changed as follows;

“The interaction of *UBAP2L* and *G3BBP1* mediated by small nucleolar RNA is required for the formation of stress granules.”

2. It is reported that *UBAP2L* can localize to both *p* body and SG. The localization of *UBAP2L* to the *p* body became more obvious in *G3BBP1/2* double KO cells, as shown in Figure S6A reported in this paper: Sanders DW, Kedersha N, Lee DSW, et al. Competing Protein-RNA Interaction Networks Control Multiphase Intracellular Organization. *Cell*. 2020;181(2):306-324.e28. doi: 10.1016/j.cell.2020.03.050

►Author Response-2

Thank you for your suggestions and comments. Sanders DW, Kedersha N, Lee DSW, et al., 2020 *Cell* reported that *UBAP2L* is found in both the SG and the *p*-body. We have also noticed that *UBAP2L*, which is found in the SG, is in contact with the *p*-body. We also believe that SG and *p*-body share proteins and functions, so the paper by Sanders DW, Kedersha N, and Lee DSW, et al., 2020 *Cell* has been cited and the text has been changed.

Because we could not obtain *G3BBP1/2* KO cells, we attempted to knock down *G3BBP1/2* with siRNA, followed by Immunofluorescence staining for DCP1A and *UBAP2L*. Confocal microscopic imaging revealed a slight increase in overlap *UBAP2L* and DCP1A in *G3BBP1/2* double knockdown cells, with a significant difference (* $P < 0.05$, Supplementary Figure.1d and h). Although the

knockdown efficiency of siRNA is considered to be weaker than that of KO cells, the localization of UBAP2L to PB is also more obvious in siRNAG3BP1/2 knock down cells. Figure 1d-h was supplemented with the results, and the following text is included in the revised manuscript.

(Page 5, lines 136-145)

“Although UBAP2L did not accumulate at DCP1A-positive dots, some UBAP2L positive dots were bordered by DCP1A positive dots and merged to a lesser extent. (Figure 1d). According to previous studies, the localization of UBAP2L to PB is more clearly visible in KO-G3BP1 and 2 cells [36], so we investigated whether UBAP2L localized to PBs in siRNAG3BP1 and 2 double transfected HeLa cells. The number of SG decreases when G3BP1 and 2 are knocked down with siRNA (Supplementary Fig.1f), but p-body formation is unaffected (Supplementary Fig.1g). A slight increase in UBAP2L localization to the PB was observed after double knockdown of G3BP1 and 2. (Supplementary Fig. 1d and h). As a result, it is proposed that UBAP2L may also work with the PBs.”

Supplementary Fig. 1

3. Figure 2A is marked as UBAP2. I think the author means UBAP2L. The same problem exists for SIC.

►Author Response-3

Thank you for pointing out the errors; we have corrected them.

4. The requirement of UBAP2L for SG formation was previously reported in the blow two papers: Youn JY, Dunham WH, Hong SJ, et al. High-Density Proximity Mapping Reveals the Subcellular Organization of mRNA-Associated Granules and Bodies. *Mol Cell*. 2018;69(3):517-532.e11. doi:10.1016/j.molcel.2017.12.020

Markmiller S, Soltanieh S, Server KL, et al. Context-Dependent and Disease-Specific Diversity in Protein Interactions within Stress Granules. *Cell*. 2018;172(3):590-604.e13. doi:10.1016/j.cell.2017.12.032

►Author Response-4

Thank you for bringing this to my attention; I have cited these previous papers and changed the text.

(Page 4, lines 110-111)

“Recent reports indicate that UBAP2L plays an important role in SGs, [31, 32, 33] but the exact function of the protein in cells is unknown.”

(Page 11, lines 327-328)

“Although there is increasing evidence that UBAP2L is required for SG formation, [31, 32, 33] the functions of mammalian UBAP2L have yet to be thoroughly investigated.”

4. A much narrower region inside UBAP2L (518-523) has been suggested to be the region responsible for G3BP interaction. This work concludes at 194-983 of UBAP2L is essential and doesn't provide novel knowledge on this interaction.

►Author Response-5

We did not create fragments of only aa518-523 or Δ aa518-523 in our deletion mutant experiment. G3BP1 did not bind to the amino acid 504-983 fragment (Supplementary Fig. 3c).

5. *How specific is G3BP1/UBAP2L binding to snoRNA since both proteins are RGG/RG containing, and those motifs are usually not specific to sequence? G3BP1 can phase separate with single-stranded RNA irrespective of sequence, indicating the binding of RNA by G3BP1 has low sequence specificity.*

►Author Response-6

The deletion mutant was used to determine which UBAP2L regions bind to snoRNA. In 293T cells, FLAG-UBAP2L, FLAG-1-193, FLAG- Δ RGG/RG (130-193), or FLAG-194-983 was expressed and precipitated with FLAG beads. RNA was extracted from each precipitate and analyzed using qRT-PCR. As illustrated in Figure 6f, FLAG- Δ RGG/RG (130-193) and FLAG-194-983 were not bound to SNORD44 and 49A, but FLAG-FL, and FLAG-1-193 were. These findings suggested that the snoRNA binding region is an RGG/RG motif. Furthermore, similar experiments on G3BP1 revealed that SNORD44 binds more to the C-terminus, including the RGG/RG and RRM region (data not shown). However, snoRNA still binds to G3BP1 even in mutants lacking RGG/RG and RRM regions, it is suggested that sites other than RGG/RG and RRM regions may also bind to snoRNA. In the future, we would like to analyze it in detail.

The result is shown in Figure 6f. The following text has changed in revised manuscript.

(Page 10, lines 302-308)

“Using deletion mutants, we then determined which regions of UBAP2L are required for snoRNA binding. In 293T cells, FLAG-UBAP2L, FLAG-1-193, FLAG- Δ RGG/RG (130-193), or FLAG-194-983 was expressed and precipitated with FLAG beads. RNA was extracted from each precipitate and analyze using qRT-PCR. As illustrated in Figure 6f, binding to snoRNA was observed in the aa1-193 region. Furthermore, the absence of the RGG/RG region, which is an RNA-binding motif, reduces binding to snoRNA. It has been proposed that it is via the RGG/RG motif of UBAP2L.”

Fig.6f

6. Are the snoRNA detected from RNA-seq mature snoRNA? Can the authors try to do FISH on the most abundant sequence detected from sequencing and check their localization to SG?

►Author Response-7

Thank you for your thoughtful observation; the sequences are mature snoRNAs. The full length of snoRNAs were cloned and used in subsequent experiments for the FISH experiment. It was difficult to assess the abundance of snoRNAs from sequence data at this point, and we would like to check them with bio-informatician as future tasks.

7. Line 333, the FISH result is Fig.S5C, not Fig.S4C.

►Author Response-8

Thank you for pointing out the errors; they have been corrected.

8. How could the author exclude the indirect role of NOP56/58 in SG assembly? Since NOP56/58 would have a dramatic effect on cell metabolism, especially nucleolus. Could the author map the RNA binding region of UBAP2L and test the RNA binding deficient mutant in SG assembly?

►Author Response-9

Thank you for pointing this out. We agree that the indirect effects of NOP56/58 need to be considered more. However, the knockdown by siRNA of NOP56/58 reduces SNORD44,49A and the binding of UBAP2L and G3BP1 (Figure 6), and this data suggests that snoRNA is involved in SG formation. As you suggested, we analyzed where the snoRNA bind to UBAP2L region using deletion mutant. We found that snoRNA binds a lot to aa1-193, including the RGG region of UBAP2L, and Δ RGG mutant was declining (Fig 6f). Unlike the binding region with G3BP1, aa1-193 is not localized to SG, but interestingly this region is localized to the nucleolus. Since aa1-193 is not localized to SG, we thought that it would not be involved in SG formation. However, when the data were carefully analyzed again, the G3BP1 binding was slightly reduced in aa194-983 compared to FL. Furthermore, in the siRNA rescue experiment of Figures 3h and 3i, cells expressing aa194-983 that binds to G3BP1 significantly rescued SG formation, but compared to cells expressing FL, the number of SG formation has slightly decreased (Figures 3h and 3i). We would like to conduct a detailed analysis as the future tasks, but it is thought that localization to SG is probably through G3BP1 binding and UBAP2L aa1-193 region is necessary for snoRNA binding. It was shown that snoRNA binding is necessary for stable binding and SG formation of UBAP2L and G3BP1 Figure 3h and 3i show the results. The revised manuscript now includes the following texts.

Fig 3

(Page 6, lines 183-185)

“However, because the aa194–983 bond is weaker than that of FL, it is possible that aa1–193 also contains the region required for G3BP1 binding (Supplementary Fig. 3c).”

(Page 7, lines 202-204)

“However, when compared to FLAG–FL–UBAP2L(R), FLAG–194–984(R) has a slight but significant difference in the number of SGs (Fig. 3h and i), indicating that FLAG–1–193 region may also play a role in SG formation.”

(Page 10, lines 302-308)

“Using deletion mutants, we then determined which regions of UBAP2L are required for snoRNA binding. In 293T cells, FLAG-UBAP2L, FLAG-1-193, FLAG-Δ RGG/RG (130-193), or FLAG-194-983 was expressed and precipitated with FLAG beads. RNA was extracted from each precipitate and analyze using qRT-PCR. As illustrated in Figure 6f, binding to snoRNA was observed in the aa1-193 region. Furthermore, the absence of the RGG/RG region, which is an RNA-binding motif, reduces binding to snoRNA. It has been proposed that it is via the RGG/RG motif of UBAP2L.”

9. *G3BP1/2 has been suggested to be the scaffold of SG formation; what role could UBAP2L play via the interaction of G3BP should be discussed. Many SG proteins are RNA binding proteins; why is UBAP2L distinct from others?*

►**Author Response-10**

We apologize for the omission of an explanation. We assumed that UBAP2L binding to other SG proteins (e.g., FXR1/2 and Caprin) would also be important in SG formation. This study found that snoRNA regulates SG formation by binding to UBAP2L and G3BP1. As shown in Figure 4f, FXR2 did not have control of binding via small RNA, so we think regulation of snoRNA is specific for G3BP1. Because many SG proteins are RNA-bound proteins, as you mentioned, snoRNA may bind to other SG proteins. In the future, we hope to conduct a thorough investigation.

We revised our conversation about this point.

(Page11, line 338-343)

“The interaction of UBAP2L with other SG proteins (for example, FXR1/2 and Caprin) may also play a role in SG formation. Our findings show that RNA does not regulate the binding of UBAP2L and FXR2, and that Halo-CPEB (SG protein that is not bound to UBAP2L) does not bind to snoRNA. Because of many SG proteins bind to RNA, some other SG proteins may bind to snoRNA, but these findings indicate that the UBAP2L/snoRNA/G3BP1 protein-RNA complex is unique.”

Reviewers' comments:

Reviewer #1 (Remarks to the Author):

In this revised manuscript, the authors main conclusion is to show that snoRNAs are the mediators of the interaction between Uba2pl and G3bp proteins that is relevant for stress granule formation. The authors present the following observations in support of this model:

- a) the co-ip of Uba2pl and G3BP1 is RNase sensitive. This data is clear, but doesn't demonstrate any role for snoRNAs.
- b) That when Uba2pl from cells is treated with RNase and immunopurified, only the addition of small RNAs can restore co-ip with recombinant G3BP1. This data is clear as presented, and implicates smaller RNAs as relevant (although a bit surprising).
- c) RNA-Seq of RNAs that co-ip with Uba2pl shows a high fraction of snoRNAs. Although it would have been important to standardize these reads to the levels of these RNAs in total small RNAs as these are very abundant RNAs. Without such standardization, it is a bit hard to interpret this result.
- d) They observe that an in vitro sense snoRD44, but not antisense, can restore co-ip of immunopurified Uba2pl with G3BP1. I found this data not very convincing since there is so little difference between the sense and antisense RNAs, and invariably these snoRNAs are heavily coated with proteins in the cell.
- e) They show that snoRNAs can be somewhat enriched in a stress granule biochemical fraction.
- and f) they show that kd of nop58 or nop56 reduces snoRNA biogenesis and can affect stress granule assembly.

Taken together, these data make a case for snoRNA dependent interactions with Uba2pl and G3bp, although there is not an unambiguous experiment that really nails this conclusion. Thus, the work could be published, and we will see if it stands the test of time and reproduction in other labs.

Reviewer #2 (Remarks to the Author):

The revised version is largely improved, but the reviewer still has some concerns as listed below.

1. It should be cautious to make the conclusion that, for instance, in the Abstract, L49-51, as well as in the Results, L134. UBAP2L is not a novel component of SGs as this has been known and the study just confirmed other's previous findings.
2. Instead, the association of UBAP2L with PB seemed more interesting, the reviewer would suggest the authors to dig this more deeply.
3. Figs 2a and 2d, after UBAP2L KD, it is not appropriate to use UBAP2L as a marker to label SGs, as UBAP2L has been knockdowned, the co-IF signals of course will be compromised. Every other two paired markers but not including UBAP2L would be more appropriate. Therefore, if UBAP2L-KD impairs the co-localization of other two markers, the conclusion is more reasonable that UBAP2L is required for SGs.
4. Please add citing the reference (Youn et al. Mol Cell. 2018; Huang et al. Cell Death Differ. 2020) when describing fig 2, as these two studies have addressed the role of UBAP2L by knocking down UBAP2L, and the study (Huang et al. Cell Death Differ. 2020) has demonstrated the interactions between UBAP2L and FXR1/2.
5. L183-184, the description seemed not correct, as the authors stated that aa 868-869 is a RRG/RG motif. Furthermore, there are also a RRM motif and a DUF motif located in aa194-869, as indicated in the reports (Youn et al. Mol Cell. 2018; Huang et al. Cell Death Differ. 2020). Discussion regarding the consistency and inconsistency of the present data to these reports are also needed.
6. L212-214 and Fig S4, Adox treatment did not affect the interaction between UBAP2L and G3BP1, this seems contradictory to the findings in Huang et al. Cell Death Differ. 2020, but the result that Adox increased SGs were concordant. This needs citing and discussion.

7. Fig 5c may lacked G3BP1 labeling.

8. The authors try to highlight the importance of snoRNA in UBAP2L/G3BP1 associations as well as in SGs. However, previous report show that snoRNA was not found in the comprehensive 283 RNA-seq analysis of the SG core fraction (Ref 41). Besides, the authors also showed that SNORD44 and SNORD49A were a slightly detected in SG (L281-282). There is neither direct evidence showing the exact existence of snoRNA (particularly SNORD44 and SNORD49A) in SGs, nor direct tool to eliminate snoRNA to investigate its role in UBAP2L/G3BP1 associations as well as in SGs. The reference (Huang et al. Cell Death Differ. 2020) has indicated that UBAP2L directly binds to G3BP1 via the DUF domain, and via the RRG/RG domain to recruit SG components including RNA and ribosomal subunits. In this regard, snoRNA may be indirectly engulfed in SGs. The present title and conclusion still seemed not solid and need more direct evidences.

Reviewer #3 (Remarks to the Author):

The authors have addressed my concerns on the experiment result. Line 217, "phosphosphatase" should be "phosphatase".

Response to the reviewer's comment.

To Reviewer 1:

We appreciate your additional comments. All points were carefully considered, and our manuscript was revised to address your comments. Please see our point-by-point responses to your comment as shown below.

Reviewer #1 (Remarks to the Author):

In this revised manuscript, the authors main conclusion is to show that snoRNAs are the mediators of the interaction between Uba2p1 and G3bp proteins that is relevant for stress granule formation. The authors present the following observations in support of this model:

a) the co-ip of Uba2p1 and G3BP1 is RNase sensitive. This data is clear, but doesnt demonstrate any role for snoRNAs.

►**Author Response-1:** It is true that our data does not show the role of snoRNA, and we would like to analyze it in the future.

b) That when Uba2p1 from cells is treated with RNase and immunopurified, only the addition of small RNAs can restore co-ip with recombinant G3BP1. This data is clear as presented, and implicates smaller RNAs as relevant (although a bit surprising).

►**Author Response-2:** Thank you for the comment. We agree that the results are surprising, so we tested repeatedly to conclude this. This insight should be considered and we would like to deal with this as a future task.

c) RNA-Seq of RNAs that co-ip with Uba2p1 shows a high fraction of snoRNAs. Although it would have been important to standardize these reads to the levels of these RNAs in total small RNAs as these are very abundant RNAs. Without such standardization, it is a bit hard to interpret this result.

►**Author Response-3:** We apologize for the confusion. We sequenced UBAP2L RIP RNA and G3BP1 RIP RNA using MiSeq, illumina. As shown in supplemental data 1 and method section "RNA sequence ", small RNA mapped using UCSC in UBAP2L RIP RNA were 1002 types and 796 in G3BP1 RIP RNA. In UBAP2L RIP RNA, the total read of 1002 types of small RNA was standardized as total 100 million reads, known as the read per million (RPM) standardization. As the result of standardizing, the small RNAs in the top 10 highest read counts contained six snoRNAs (As shown in Fig5a). In addition, looking at the top 100, 51% of RNAs were categorized into snoRNA. From the above, we interpreted that snoRNA is abundant in small RNA fraction.

d) They observe that an in vitro sense snoRD44, but not antisense, can restore co-ip of immunopurified Uba2pl with G3BP1. I found this data not very convincing since there is so little difference between the sense and antisense RNAs, and invariably these snoRNAs are heavily coated with proteins in the cell.

►**Author Response-4:** Thank you for pointing it out. We totally agree that this data is not perfect to convince our claims. To address this point, we created each RNA in vitro translation. To synthesizing in vitro, the snoRNA sequence is cloned into plasmid, and the plasmid promoter is used to translation. Therefore, the RNA created for this experiment also contains a plasmid sequence. Due to the low number of bases in snoRNA (ex. SNORD44 is 61base pair RNA), sensibility may have been low due to extra RNA, such as these plasmid sequences. However, in our experiments, a difference was observed between sense and anti-sense, and a quantitative about decrease of 75% was obtained when ImageJ was used. At least, this data supports our conclusion. We understand the point should be clarified by testing other assays to prove. However, we would like to work on this as next project.

We show the quantitative values of WB in Figure 5d and the following text is added in the Figure 5d legend.

(Page25 Line795-Page26 Line 796)

The number in the middle of the figure and the graphs indicates the quantitative value measured with ImageJ.

Figure 5d

e) They show that snoRNAs can be somewhat enriched in a stress granule biochemical fraction. and f) they show that kd of nop58 or nop56 reduces snoRNA biogenesis and can affect stress granule assembly.

Taken together, these data make a case for snoRNA dependent interactions with Uba2p1 and G3bp, although there is not an unambiguous experiment that really nails this conclusion. Thus, the work could be published, and we will see if it stands the test of time and reproduction in other labs.

►**Author Response-5:** Thanks for your comment. In this study, we showed that snoRNA plays an important role in the SGs, and also demonstrated the relationship between UBAP2L, G3BP1 and snoRNA. However, we regret that we have not yet shown an experiment that gets to the core. In the future, as you pointed out, we would like to continue the analysis and show more reliable proof.

To Reviewer 2:

We would like to thank the reviewer for many insightful comments that helped us improve our manuscript. Please see the following for our point-by-point responses to your comment.

Reviewer #2 (Remarks to the Author):

The revised version is largely improved, but the reviewer still has some concerns as listed below.

1. It should be cautious to make the conclusion that, for instance, in the Abstract, L49-51, as well as in the Results, L134. UBAP2L is not a novel component of SGs as this has been known and the study just confirmed other's previous findings.

►**Author Response-1:** Thank you for point-out. We revised the Abstract and Results explanations and cited the pioneer works.

(Page2 Line 48-49)

In this report, we show that ubiquitin-associated protein 2-like (UBAP2L) is a crucial component of SGs.

(Page2 Line55-57)

Our results reveal a critical role of an SG component, the UBAP2L/snoRNA/G3BP1 protein-RNA complex, and provide new insights into the regulation of SG assembly.

(Page 4 Line 119-121)

As shown in Figure 1a, UBAP2L accumulated into multiple small dot-like structures and colocalized with each of these marker proteins 30 min after arsenite or sorbitol treatment, as in previous reports [31,32,33,34].

(Page5 Line 134-135)

These results clearly show that UBAP2L is an essential component of SGs.

(Page10 Line315)

In this report, we identified a crucial role for UBAP2L in SG organization.

(Page10 Line319-320)

These results clearly show that UBAP2L is a crucial component of SGs and essential for SG assembly.

2. Instead, the association of UBAP2L with PB seemed more interesting, the reviewer would suggest the authors to dig this more deeply.

►**Author Response-2:** We appreciate your comment, and agree that the point is interesting. As previously reported, UBAP2L is a protein localized to both SG and PB, and is a protein that mediates between SG and PB. In the future, we would like to analyze the role of UBAP2L in PB.

3. Figs 2a and 2d, after UBAP2L KD, it is not appropriate to use UBAP2L as a marker to label SGs, as UBAP2L has been knocked down, the co-IF signals of course will be compromised. Every other two paired markers but not including UBAP2L would be more appropriate. Therefore, if UBAP2L-KD impairs the co-localization of other two markers, the conclusion is more reasonable that UBAP2L is required for SGs.

►**Author Response-3:** Thank you for the suggestion. We apologize for the lack of explanation. We immunostained UBAP2L in this experiment because we wanted to observe G3BP1 and other SG marker signals in cells with UBAP2L-KD.

4. Please add citing the reference (Youn et al. Mol Cell. 2018; Huang et al. Cell Death Differ. 2020) when describing fig 2, as these two studies have addressed the role of UBAP2L by knocking down UBAP2L, and the study (Huang et al. Cell Death Differ. 2020) has demonstrated the interactions between UBAP2L and FXR1/2.

►**Author Response-4:** Thank you for the point out. We agree that these reports will help us in our work. The prior works were included in the revised manuscript as references, and the following sentences were added.

(Page4 Line 110-111)

Recent evidence suggests that UBAP2L plays an important role in SGs [31, 32, 33, 34]; but, the exact function of the protein in cells is unknown.

(Page4 Line 119-121)

As shown in Figure 1a, UBAP2L accumulated into multiple small dot-like structures and colocalized with each of these marker proteins 30 min after arsenite or sorbitol treatment, as in previous reports [31,32,33,34].

(Page5 Line150-152)

As shown in Figure 2a, transfection of UBAP2L siRNAs disrupted SG organization. This result is consistent with previous reports [33, 34]

(Page6 Line167-169)

We performed mass spectrometry analysis to search for proteins which associate with UBAP2L. In this analysis, G3BP1/2, Caprin1 and the fragile X mental retardation family proteins; FXR1/2 and FMRP, which are SG-localized proteins [18–20,33,34,38]

5. L183-184, the description seemed not correct, as the authors stated that aa 868-869 is a RRG/RG motif. Furthermore, there are also a RRM motif and a DUF motif located in aa194-869, as indicated in the reports (Youn et al. Mol Cell. 2018; Huang et al. Cell Death Differ. 2020). Discussion regarding the consistence and inconsistency of the present data to these reports are also needed.

►**Author Response-5:** Thanks for pointing it out. RNA binding and DUF motif have been described, the following text has been added in the Result section.

(Page6 Line179-181)

UBAP2L has a ubiquitin-associated domain (aa50–80), RGG/RG motifs (aa134–189), the two putative RNA binding region (aa239–290), and DUF (domain unknown function) motif (aa 495–526).

(Page6 Line184-186)

We found that the aa194–983 fragment of UBAP2L, which contains the DUF motif, was required for the interaction with G3BP1, in line with previous findings [34].

6. L212-214 and Fig S4, Adox treatment did not affect the interaction between UBAP2L and G3BP1, this seem contradictory to the findings in Huang et al. Cell Death Differ. 2020, but the result that Adox increased SGs were concordant. This need citing and discussion.

►**Author Response-6:** Thank you for the suggestion. We re-conducted experiments to check SG formation by Adox, but we did not obtain similar results as Huang C et al (Figure for reviewers 1). As shown in the Figure for reviewers 2, we also experimented with PRMT1 siRNA to see if SG formation increased, but no change was observed in SG formation. Many factors can affect the results and the further analysis seems to be needed to conclude this.

In the revised manuscript, we cited the paper to further discuss this point, and the following texts in Discussion section.

Figure for reviewers 1.

The cells were cultured with or without Adox for 24 h, and then treated with 0.5 mM arsenite for 30 min. The cells were immunostained for G3BP1 and UBAP2L. The numbers of SGs positive for G3BP1 per cell are presented in the graph. Three independent experiments were performed and 10 cells were evaluated for each experiment (N.S. not significant).

Figure for reviewers 2.

The cells were transfected with control or PRMT1 siRNAs, and 72 h later, the cells were treated with 0.5 mM arsenite for 30 min and immunostained with anti-G3BP1 antibodies. The numbers of SGs positive for G3BP1 per cell are presented in the graph. Three independent experiments were performed and 10 cells were evaluated for each experiment. (N.S. not significant)

(Page11 Line336-339)

Furthermore, UBAP2L is known to undergo methylation by PRMT1 [26,34]. Huang C et al reported that Adox treatment increases SG formation [34]. Our results were not perfectly concordant with this report, and there might be complicated processes. Further analysis is needed to clarify this point.

7. Fig 5c may lacked G3BP1 labeling.

►**Author Response-7:** Thank you for pointing out the errors; we have corrected them.

8. The authors try to highlight the importance of snoRNA in UBAP2L/G3BP1 associations as well as in SGs. However, previous reports show that snoRNA was not found in the comprehensive 283 RNA-seq analysis of the SG core fraction (Ref 41). Besides, the authors also showed that SNORD44 and SNORD49A were a slightly detected in SG (L281-282). There is neither direct evidence showing the exact existence of snoRNA (particularly SNORD44 and SNORD49A) in SGs, nor direct tool to eliminate snoRNA to investigate its role in UBAP2L/G3BP1 associations as well as in SGs. The reference (Huang et al. Cell Death Differ. 2020) has indicated that UBAP2L directly binds to G3BP1 via the DUF domain, and via the RRG/RG domain to recruit SG components including RNA and ribosomal subunits. In this regard, snoRNA may be indirectly engulfed in SGs. The present title and conclusion still seemed not solid and need more direct evidences.

►**Author Response-8:** Thank you for insightful comment. We are aware that we have not yet been able to conduct a perfect experiment on the involvement of snoRNA in SG formation. However, our experimental results suggest that snoRNA is involved in SG formation by mediating the binding of UBAP2L and G3BP1, and regarding this point, we think it is a new insight. We would like to further analyze whether snoRNA is directly involved in SG formation as a future work.

To Reviewer 3:

We would like to thank the reviewer for the comments that helped us improve our manuscript.

Reviewer #3 (Remarks to the Author):

The authors have addressed my concerns on the experiment result. Line 217, "phosphosphatase" should be "phosphatase".

►**Author Response:** Thank you for pointing out the errors; we have corrected them.